# STRAP: Robot Sub-Trajectory Retrieval for Augmented Policy Learning

**Marius Memmel[1,*], Jacob Berg[1,*], Bingqing Chen[2], Abhishek Gupta[1,†], Jonathan Francis[2,3,†]**

[1]Paul G. Allen School of Computer Science & Engineering, University of Washington
[2]Robot Learning Lab, Bosch Center for Artificial Intelligence
[3]Robotics Institute, Carnegie Mellon University
{memmelma,jacob33,abhgupta}@cs.washington.edu,
{bingqing.chen,jon.francis}@us.bosch.com

## ABSTRACT

Robot learning is witnessing a significant increase in the size, diversity, and complexity of pre-collected datasets, mirroring trends in domains such as natural language processing and computer vision. Many robot learning methods treat such datasets as multi-task expert data and learn a multi-task, generalist policy by training broadly across them. Notably, while these generalist policies can improve the average performance across many tasks, the performance of generalist policies on any one task is often suboptimal due to negative transfer between partitions of the data, compared to task-specific specialist policies. In this work, we argue for the paradigm of training policies during deployment given the scenarios they encounter: rather than deploying pre-trained policies to unseen problems in a zero-shot manner, we non-parametrically retrieve and train models directly on relevant data at test time. Furthermore, we show that many robotics tasks share considerable amounts of low-level behaviors and that retrieval at the *"sub"-trajectory* granularity enables significantly improved data utilization, generalization, and robustness in adapting policies to novel problems. In contrast, existing full-trajectory retrieval methods tend to underutilize the data and miss out on shared cross-task content. This work proposes STRAP, a technique for leveraging pre-trained vision foundation models and dynamic time warping to retrieve sub-sequences of trajectories from large training corpora in a robust fashion. STRAP outperforms both prior retrieval algorithms and multi-task learning methods in simulated and real experiments, showing the ability to scale to much larger offline datasets in the real world as well as the ability to learn robust control policies with just a handful of real-world demonstrations. Project website at https://weirdlabuw.github.io/strap/

## 1 INTRODUCTION

Robot learning techniques have shown the ability to shift the process of designing robot controllers from a large manual or model-based process to a data-driven one (Francis et al., 2022; Hu et al., 2023). Especially, end-to-end imitation learning with, *e.g.*, diffusion models (Chi et al., 2023; Wang et al., 2024) and transformers (Haldar et al., 2024), have shown impressive success. While imitation learning can be effective for

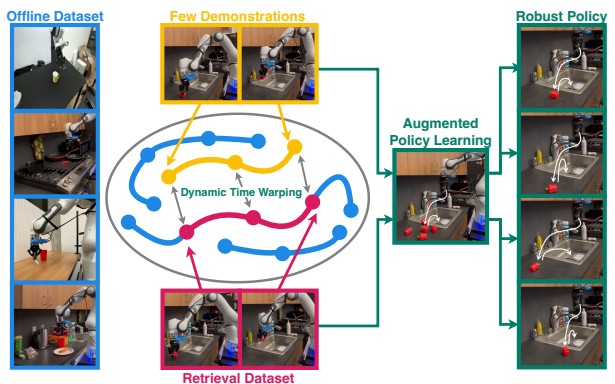

Figure 1: **STRAP** retrieves sub-trajectories using subsequence dynamic time warping for training robust policies during deployment.

---

*equal contribution † equal advising

performing particular tasks with targeted in-domain data collection, this process can be expensive and time-consuming in terms of human effort. This becomes a challenge as we deploy robots into dynamic environments such as homes and offices, where new tasks and environments are commonplace and constant data collection is impractical.

Multi-task policy learning is often applied in such situations, where data across multiple tasks is used to train a large task- or instruction-conditioned model that has the potential to generalize to new problems. While multi-task learning has seen successes in certain settings (Reed et al., 2022; Brohan et al., 2022), the performance of a multi-task, generalist policy is often lower than task-specific, specialist policies. This can be attributed to the model suffering from negative transfer and sacrificing per-task performance to improve the average performance across tasks. This challenge is exacerbated in unseen tasks or domains since zero-shot generalization is challenging and collecting large amounts of in-domain fine-tuning data can be expensive. In this work, we consider training expert models during test time as a better way to use pre-collected datasets and enable few-shot imitation learning for new tasks.

In particular, we build on the paradigm of *non-parametric data retrieval*, where a small amount of in-domain data collected at test-time is used to retrieve a subset of particularly "relevant" data from the training corpus. This retrieved data can then be used for robust and performant model training on new tasks. In this sense, the retrieved data can guide learned models towards desired behavior; however, the question becomes: *How do we sub-select which data to retrieve from a large, pre-existing corpus?*

Several prior techniques have studied the problem of non-parametric retrieval from the perspective of learning latent embeddings that encode states (Du et al., 2023), skills (Nasiriany et al., 2022), optical flow (Lin et al., 2024), and learned affordances (Kuang et al., 2024). Most techniques are challenging to apply out of the box for two primary reasons. Firstly, they require training domain-specific encoders to embed states, skills, or affordances: this makes it challenging to apply to demonstrations collected in the open world, where visual appearance can show wide variations. Secondly, they often retrieve entire trajectories, limiting the policies' ability to use data from other tasks that may share common components with the desired test-time behavior. These challenges limit both the broad applicability of these retrieval methods and the amount of cross-task data sharing. *How can we design easy-to-use off-the-shelf retrieval methods that maximally utilize the training data for test-time adaptation?*

The key insight in this work is that retrieval methods do not need to measure the similarity between entire trajectories (or individual states), but rather between *sub-trajectories* of the desired behavior at test-time and corresponding sub-trajectories of the training data. Notably, these sub-trajectories do not need to come from tasks that are similar in entirety to the desired test-time tasks. Instead, sub-components of many related tasks can be shared to enable robust, test-time policy training. For example, as shown in Fig. 1, for the multi-stage task of *"pick up the mug, put it in the drawer, and close it"*, both *"pick up the mug*, put in on top of the drawer" and *"close the bottom drawer*, open the top drawer" contain sub-tasks that when retrieved provide useful training data. Our proposed method, **S**ub-sequence **T**rajectory **R**etrieval for **A**ugmented **P**olicy Learning (STRAP), uses a small amount of in-domain trajectories collected at test-time to retrieve and train on these relevant sub-trajectories across a large multi-task training corpus. The resulting policies show considerable improvements in robustness and generalization over previous retrieval methods, zero-shot multi-task policies, or policies that are trained purely on test-time in-domain data.

We show how STRAP can be used with minimal effort across training and evaluation domains with non-trivial visual differences. Our method first compares sub-trajectory similarity using features from off-the-shelf foundation models, *e.g.*, DINOv2 (Oquab et al., 2023); these features capture strong notions of "object-ness", discarding spurious visual differences such as lighting, texture, and local changes in object appearance. Secondly, our method leverages time-invariant alignment techniques, such as dynamic time warping (Giorgino, 2009), to compute the similarity between sub-trajectories of different lengths, removing requirements for retrieved trajectories to have a similar length and increasing the applicability of STRAP across tasks and domains. Lastly, we show how STRAP can be applied to arbitrary test corpora, with sub-trajectories being automatically extracted by our framework, thereby removing the requirement for manual segmentation of relevant sub-trajectories from the training corpus. We demonstrate how STRAP can be used out of the box to augment *any* few-shot imitation learning algorithm, providing significant gains in generalization

at test-time, while avoiding expensive, test-time in-domain data collection. We instantiate STRAP with transformer-based imitation learning policies and show the benefits of few-shot sub-trajectory retrieval on the LIBERO (Liu et al., 2024) benchmark in simulation and real-world imitation learning problems.

## 2 RELATED WORK

**Retrieval for Behavior Replay:** A considerable body of work has explored retrieval-based approaches for robotic manipulation, where the retrieval of relevant past demonstrations aids in replaying past experiences. The choices of embedding space hereby range from off-the-shelf models (Di Palo & Johns, 2024; Malato et al., 2024) like DINO (Caron et al., 2021), training encoders on the offline dataset (Pari et al., 2022) to abstract representation like object shapes (Sheikh et al., 2023). Some works do not directly replay actions but add a layer of abstraction following subgoals (Zhang et al., 2024b), affordances (Kuang et al., 2024) or keypoints (Papagiannis et al., 2024). A key assumption of these methods is that the offline data either exactly resembles expert demonstrations collected in the test environment or that intermediate representations can bridge the gap. These drawbacks limit the usage of large multi-task datasets collected in various domains.

**Retrieval for Few-shot Imitation Learning:** Retrieval for policy learning tries to mitigate these issues by learning policies from the retrieved data. While retrieval has shown to benefit policy learning from sub-optimal single-task data (Yin & Abbeel, 2024), most work focuses on retrieving from large multi-task datasets like DROID (Khazatsky et al., 2024) or OpenX (O'Neill et al., 2023) containing expert demonstrations. BehaviorRetrieval (BR) (Du et al., 2023) and FlowRetrieval (FR) (Lin et al., 2024) train an encoder-decoder model on state-action and optical flow respectively. Related to our work, SAILOR (Nasiriany et al., 2022) imposes skill constraints on the embedding space, clustering similar skills together to later retrieve those. A significant downside of training custom representations is that these methods do not scale well to the increasing size of available offline datasets and are unable to deal with significant visual and semantic differences. Moreover, techniques like BehaviorRetrieval and FlowRetrieval retrieve individual states, rather than sub-trajectories like our work, where sub-trajectory retrieval enables maximal data sharing between seemingly different tasks while capturing temporal information.

**Learning from Sub-trajectories:** Several works propose to decompose demonstrations into reusable sub-trajectories, e.g., based on end-effector-centric or full proprioceptive state-action transitions (Li et al., 2020; Belkhale et al., 2024; Shankar et al., 2022; Myers et al., 2024; Francis et al., 2022). Belkhale et al. (2024) propose to decompose demonstrations into end-effector-centric subtasks, *e.g.*, "move forward" or "rotate left". The authors show that by decomposing and re-labeling the language instructions into a shared vocabulary, knowledge from multi-task datasets can be better shared when training multi-task policies. Myers et al. (2024) leverage VLMs to decompose demonstrations into sub-trajectories to better learn to imitate them. To our knowledge, we propose the first robot sub-trajectory retrieval mechanism, for partitioning large offline robotics datasets and for enabling cross-task positive transfer during policy learning.

## 3 PRELIMINARIES

### 3.1 DYNAMIC TIME WARPING

To match sequences of potentially variable length during retrieval, we build on an algorithm called dynamic time warping (DTW) (Müller, 2021). DTW methods compute the similarity between two time series that may vary in time or speed, *e.g.*, different video or audio sequences. This algorithm aligns the varying length sequences by warping the time axis of the series using a set of step sizes to minimize the distance between corresponding points while obeying boundary conditions.

DTW algorithms are given two sequences, $X = \{x_1, x_2, \ldots, x_n\}$ and $Y = \{y_1, y_2, \ldots, y_m\}$, where $m \neq n$, and a corresponding cost matrix $C(x_i, y_j)$ that assigns the cost of assigning element $x_i$ of sequence $X$ to correspond with element $y_j$ of sequence $Y$. The goal of DTW is to find a mapping between $X$ and $Y$ that minimizes the total cumulative distance between the assigned elements of both sequences while obeying boundary and continuity conditions. Dynamic time warping methods solve this problem efficiently using dynamic programming methods.

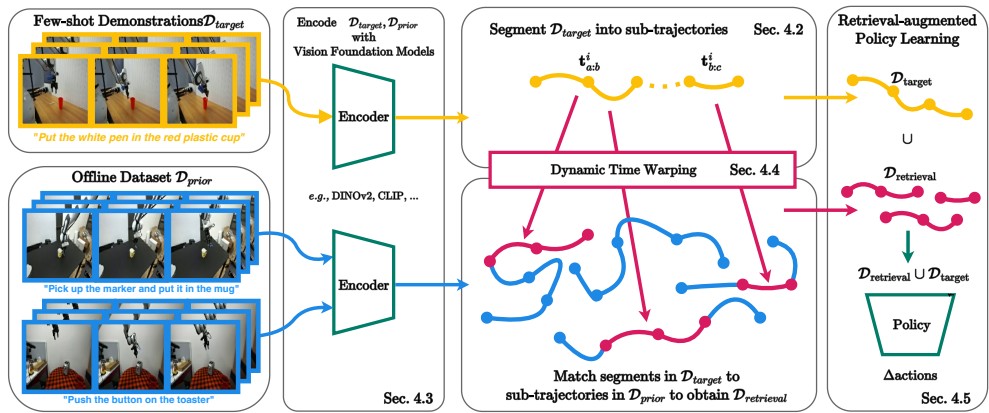

Figure 2: **Overview of STRAP:** 1) demonstrations $\mathcal{D}_{\text{target}}$ and offline datasets $\mathcal{D}_{\text{prior}}$ are encoded into a shared embedding space using a vision foundation model, 2) automatic slicing generates sub-trajectories which 3) S-DTW matches to corresponding sub-trajectories in $\mathcal{D}_{\text{prior}}$ creating $\mathcal{D}_{\text{retrieval}}$, 4) training a policy on the union of $\mathcal{D}_{\text{retrieval}}$ and $\mathcal{D}_{\text{target}}$ results in better performance and robustness.

A cumulative distance matrix $D$ is computed via dynamic programming as follows: $D(0,0) = C(0,0), D(n,1) = \sum_{k=1}^{n} C(k,1)$ for $n \in [1:N]$ and $D(1,m) = \sum_{k=1}^{m} C(1,k)$ for $m \in [1:M]$. Then the following dynamic programming calculation is performed:

$$D(i,j) = C(x_i, y_j) + \min\{D(i-1,j), D(i,j-1), D(i-1,j-1)\}, \tag{1}$$

where $C(x_i, y_j)$ is the distance between points $x_i$ and $y_j$. We assume this cost matrix is pre-provided, and we describe how we compute this from raw camera images in Sec. 4.3. The optimal alignment between the sequences is found by backtracking from $D(n,m)$ to $D(0,0)$. This guarantees that the start is matched to the start and the end is matched to the end or that the pairs $(x_0, y_0)$ and $(x_n, y_m)$ are the start and end of the path. This optimal paring path consists of the best possible alignment between $X$ and $Y$ such that the cumulative cost between all matched pairs is minimized. DTW, as described, is widely used in time-series analysis, speech recognition, and other domains where temporal variations exist between sequences. In the context of our retrieval problem, DTW is used to go beyond retrieving exactly matched sequences to matching variable length subsequences, as we describe below.

**Subsequence dynamic time warping (S-DTW)** is an extension of the DTW algorithm for scenarios where a shorter query sequence must be matched to a portion of a longer reference sequence. Given a query sequence $X = \{x_1, x_2, \ldots, x_n\}$ and a much longer reference sequence $Y = \{y_1, y_2, \ldots, y_m\}$, the goal of S-DTW is to find a subsequence of $Y$ (of a potentially different length from $X$), denoted $Y_{i:j}$ where $i \leq j$, that has the minimal DTW distance to $X$.

The cumulative cost matrix $D$ for S-DTW is computed similarly to the traditional DTW described above but allows alignment to start and end at any point in $R$. D is initialized as

$$D(0,0) = C(0,0),$$
$$D(n,1) = \sum_{k=1}^{n} C(k,1) \quad \text{for } n \in [1:N],$$
$$D(1,m) = C(1,m) \text{ for } m \in [1:M]$$

and then completed using dynamic programming following Eq. (1). This ensures that the query can match any sub-sequence of the reference. Once the cumulative cost matrix is computed, the optimal alignment is found by backtracking from the minimal value in the last row of the matrix, i.e., $\min(D(n,j))$ for $j \in \{1, \ldots, m\}$. This gives the subsequence of $Y$ that best aligns with $X$, obeying only temporality while relaxing the boundary condition. As we will show, using S-DTW for data retrieval enables the maximal retrieval of data across tasks in a retrieval-augmented policy training setting, as described in Sec. 4.3.

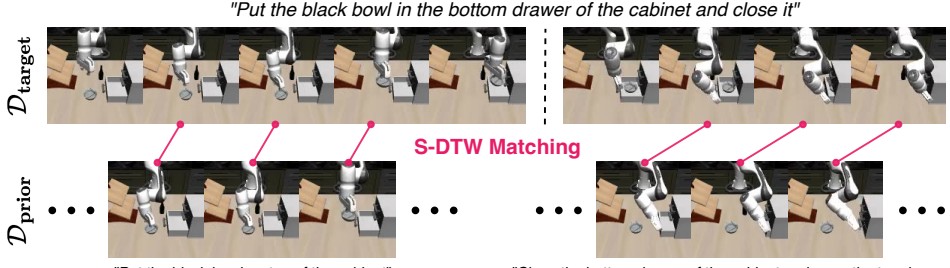

Figure 3: **Sub-trajectory matching:** S-DTW matches the sub-trajectories of $\mathcal{D}_{\text{target}}$ (top) to the relevant segments in $\mathcal{D}_{\text{prior}}$. A feature of S-DTW is that the start and end of the trajectories do not have to align, finding optimal matches for each pairing.

# 4  STRAP: SUB-SEQUENCE ROBOT TRAJECTORY RETRIEVAL FOR AUGMENTED POLICY TRAINING

## 4.1  PROBLEM SETTING: RETRIEVAL-AUGMENTED POLICY LEARNING

We consider a few-shot learning setting where we're given a target dataset $\mathcal{D}_{\text{target}} = \{(s_0^i, a_0^i, s_1^i, a_1^i, \ldots, s_{H_i}^i, a_{H_i}^i, l^i)\}_{i=1}^N$ containing expert trajectories of states $s$ (*e.g.*, observations like camera views $o$ and proprioception $x$), actions $a$ (such as robot controls), and task-specifying language instructions $l$. This target dataset is collected in the test environment and task, but there is only a small set of $N$ trajectories, which limits generalization for models trained purely on such a small dataset. Since $\mathcal{D}_{\text{target}}$ is often insufficient to solve the task alone, we posit that generalization can be accomplished by non-parametrically *retrieving* data from an offline dataset $\mathcal{D}_{\text{prior}}$. This offline dataset $\mathcal{D}_{\text{prior}} = \{(s_0^j, a_0^j, s_1^j, a_1^j, \ldots, s_{H_j}^j, a_{H_j}^j, l^j)\}_{j=1}^M$ can contain data from different environments, scenes, levels of expertise, tasks, or embodiments. Notably, the set of tasks in the offline dataset do *not* need to overlap with the set of tasks in the target dataset. We assume that the offline dataset shares matching embodiment with the target dataset and consists of expert-level trajectories, but may consist of a diversity of scenes and tasks that vary widely from the target dataset $\mathcal{D}_{\text{target}}$.

Given $\mathcal{D}_{\text{prior}}$ and $\mathcal{D}_{\text{target}}$, the goal is to learn a language-conditioned policy $\pi_\theta(a|s, l)$ that can predict optimal actions $a$ in the target environment when prompted with the current state $s$ and language instruction $l$. Assuming we can obtain a measure of success (such as task completion), and a broad set of initial conditions $s_0 \sim \rho_{\text{test}}(s_0)$ in the test environment. The objective of policy learning is to determine the policy parameters $\theta$ to maximize the expected success metric when evaluated on test conditions, under the policy $\pi_\theta$ and test-time environment dynamics. Since we are only provided a limited corpus of data, $\mathcal{D}_{\text{target}}$, in the target domain, these policy parameters cannot be learned by simply performing maximum likelihood on $\mathcal{D}_{\text{target}}$. Instead, we will present an approach where a smaller, "relevant" subset of the offline dataset $\mathcal{D}_{\text{retrieval}} \subseteq \mathcal{D}_{\text{prior}}$ is retrieved non-parametrically and then mixed with the smaller in-domain dataset $\mathcal{D}_{\text{target}}$ to construct a larger, augmented training dataset, *i.e.*, $\mathcal{D}_{\text{target}} \cup \mathcal{D}_{\text{retrieval}}$, which is most relevant to the desired test-time conditions $\rho_{\text{test}}(s_0)$. This can then be used for training policies via imitation learning, as we will describe in Sec. 4.5. Doing so avoids an expensive generalist training procedure and rather focuses the learned model to being a high-performing specialist in a particular setting of interest. The key questions becomes - *How can we define what subset of the offline dataset $\mathcal{D}_{prior}$ is relevant to construct $\mathcal{D}_{retrieval}$?*

To handle the unique nature of robotic data, e.g., multi-modal and temporally dependent, we design STRAP for retrieval-augmented policy learning. Firstly, we need to define the unit of retrieval. Rather than retrieving individual state-action pairs or entire trajectories, STRAP crucially retrieves sub-trajectories. We also propose a method to automatically segment trajectories in $\mathcal{D}_{\text{target}}$ into such sub-trajectories (Sec. 4.2). Secondly, we need to define a suitable distance metric for a pair of sub-trajectories (Sec. 4.3). Then, we need a computationally efficient algorithm to retrieve relevant sub-trajectories non-parametrically from the training set (Sec. 4.4). Finally, we put everything together and train policies based on retrieved data (Sec. 4.5).

## 4.2 SUB-TRAJECTORIES FOR DATA RETRIEVAL

To make the best use of the training dataset while capturing temporal task-specific dynamics, we expand the notion of retrieval from being able to retrieve entire trajectories or single states to retrieving variable-length sub-trajectories. In doing so, retrieval can capture the temporal dynamics of the task, while still being able to share data between seemingly different tasks with potentially different task instruction labels. We define a sub-trajectory as a consecutive subset of a trajectory $\boldsymbol{t}_{a:b}^i \subseteq T^i$ with the sub-trajectory $\boldsymbol{t}_{a:b}^i = (s_a^i, s_{a+1}^i, \ldots, s_b^i)$ including timestep $a$ to $b$ of the whole trajectory $T^i$ of length $H_i$. Most long-horizon problems observed in robotics datasets (Liu et al., 2024; Khazatsky et al., 2024; O'Neill et al., 2023) naturally contain multiple such sub-trajectories. For instance, the task shown in Fig. 3 can be decomposed into "put the bowl in the drawer" and "close the drawer". Note that we do not require these trajectories to explicitly have a specific semantic meaning, but semantically meaningful sub-trajectories often coincide with those most commonly encountered across tasks as we see in our experimental evaluation.

Given this definition of a sub-trajectory, our proposed retrieval technique only requires segmenting the target demonstrations into sub-trajectories $\mathcal{T}_{\text{target}} = \{\boldsymbol{t}_{1:a}^i, \boldsymbol{t}_{a:b}^i, \ldots, \boldsymbol{t}_{H_i-p_i:H_i}^i, \forall T^i \in \mathcal{D}_{\text{target}}\}$ but *not* the much larger offline training dataset $\mathcal{D}_{\text{prior}}$. Instead, appropriate sub-sequences will be retrieved from this dataset using a DTW based retrieval algorithm (Sec. 4.4). This makes the proposed methodology far more practical since $\mathcal{D}_{\text{prior}}$ is much larger than $\mathcal{D}_{\text{target}}$. While this separation into sub-trajectories can be done manually during data collection, we propose an automatic technique for sub-trajectory separation that yields promising empirical results. Building on techniques proposed by Belkhale et al. (2024), we split the demonstrations into atomic chunks, *i.e.*, lower-level motions, before retrieving similar sub-trajectories with our matching procedure (Sec. 4.4). In particular, we propose a simple proprioception-based segmentation technique that optimizes for changes in the robot's end-effector motion indicating the transition between two chunks. For example, a Pick&Place task can be split into picking and placing separated by a short pause when grasping the object. Let $x_t$ be a vector describing the end-effector position at timestep $t$. We define "transition states" where the absolute velocity drops below a threshold: $\|\dot{x}\| < \epsilon$ [1]. We empirically find that this proprioception-driven segmentation can perform reasonable temporal segmentation of target trajectories into sub-components. This procedure can certainly be improved further via techniques in action recognition using vision-foundation models (Team et al., 2023; Zhang et al., 2024a), or information-theoretic segmentation methods (Jiang et al., 2022).

## 4.3 FOUNDATION MODEL-DRIVEN RELEVANCE METRICS FOR RETRIEVAL

Given the definition and automatic segmentation of sub-trajectories, we must define a measure of similarity that allows for the retrieval of appropriate *relevant* sub-trajectory data from $\mathcal{D}_{\text{prior}}$, and at the same time is robust to variations in visual appearance, distractors, and irrelevant spurious features. While prior work has suggested objectives to train such similarity metrics through representation learning (Du et al., 2023; Lin et al., 2024; Kuang et al., 2024), these methods are often trained purely in-domain, making them particularly sensitive to aforementioned variations. While using more lossy similarity metrics based on optical flow (*c.f.* (Lin et al., 2024)) or language (Zha et al., 2024) can help with this fragility, it often fails to capture the necessary task-specific or semantic details. This suggests the need for a robust, domain-agnostic similarity metric that can easily be applied out-of-the-box.

In this work, we will adopt the insight that vision(-language) foundation models (Oquab et al., 2023; Radford et al., 2021) offer off-the-shelf solutions to this problem of measuring the semantic and visual similarities between sub-trajectories, capturing object- and task-centric affordances, while being robust to low-level variations in scene appearance. Trained on web-scale real-world image(-text) data, these models are typically robust to low-level perceptual variations, while providing semantically rich representations that naturally capture a notion of object-ness and semantic correspondence. Denoting a vision foundation model as $\mathcal{F}(\cdot)$, we can compute the pairwise distance of two camera views with an L2 norm[2] in embedding space, *i.e.*, $\|\mathcal{F}(o_i) - \mathcal{F}(o_j)\|_2$. While aggregation methods such as temporal averaging could be used to go from embedding of a single image

---

[1] For trajectories involving "stop-motion", this heuristic returns many short chunks as the end-effector idles, waiting for the gripper to close. To ensure a minimum length, we merge neighboring chunks until all are $\geq 20$.

[2] Other cost metrics such as (1-cosine similarity) could be used here as well.

---

**Algorithm 1** STRAP ($\mathcal{D}_{\text{target}}, \mathcal{D}_{\text{prior}}, K, \epsilon, \mathcal{F}$)

---

**Require:** demos $\mathcal{D}_{\text{target}}$, offline dataset $\mathcal{D}_{\text{prior}}$, vision foundation model $\mathcal{F}$, # retrieved chunks $K$, chunking threshold $\epsilon$;
1: /* Pre-processing */
2: $\mathcal{T}_{\text{target}} \leftarrow \texttt{SubTrajSegmentation}(\mathcal{D}_{\text{target}}, \epsilon)$;                   ▷ Heuristic demo chunking
3: $\mathcal{E}_{\text{prior}} \leftarrow \{\{\mathcal{F}(o_t)|o_t \in T\}|T \in \mathcal{D}_{\text{prior}}\}$;                    ▷ Embed $\mathcal{D}_{\text{prior}}$
4: $\mathcal{E}_{\text{target}} \leftarrow \{\{\mathcal{F}(o_t)|o_t \in T\}|T \in \mathcal{T}_{\text{target}}\}$;                 ▷ Embed chunked $\mathcal{D}_{\text{target}}$
5: /* Sub-trajectory Retrieval using S-DTW*/
6: **for** $\mathbf{S}_{\text{target}} \in \mathcal{D}_{\text{target}}$ **do**
7:     $\mathcal{M} \leftarrow []$;                                          ▷ Initialize empty match storage
8:     **for** $T_{\text{prior}} \in \mathcal{D}_{\text{prior}}$ **do**
9:         $D \leftarrow \texttt{computeCostMatrix}(\mathcal{E}_{\text{target}}, \mathcal{E}_{\text{prior}})$;            ▷ Eq. (2)
10:        $\mathcal{M}_{i,j} \leftarrow \texttt{extractSubTrajectory}(D, T_{\text{prior}})$;     ▷ Dynamic Programming
11:    **end for**
12: **end for**
13: $\mathcal{D}_{\text{retrieval}} \leftarrow \texttt{retrieveTopKMatches}(\mathcal{M}, K)$;                    ▷ Sec. 4.4
14: /* Policy Learning */
15: **repeat**
16:    sample $\mathcal{B} \sim \mathcal{D}_{\text{target}} \cup \mathcal{D}_{\text{retrieval}}$ to update policy $\pi_\theta$ with loss $\mathcal{L}(\mathcal{B}; \theta)$              ▷ Eq. (3)
17: **until** $\pi_\theta$ converged; **return** $\pi_\theta$

---

to that of a sub-trajectory, they lose out on the actions and dynamics. We instead opt for a sub-trajectory matching procedure based on the idea of DTW (Giorgino, 2009) and use the embeddings for finding maximally relevant sub-trajectories. Given two sub-trajectories, $\boldsymbol{t}_i$ and $\boldsymbol{t}_j$, we compute a pairwise cost matrix $C \in \mathbb{R}^{|\boldsymbol{t}_i| \times |\boldsymbol{t}_j|}$, where its value is as computed by:

$$C(i, j) = ||\mathcal{F}(o_i) - \mathcal{F}(o_j)||_2 \tag{2}$$

## 4.4 ᴜʙ-ᴛʀᴀᴊᴇᴄᴛᴏʀʏ Rᴇᴛʀɪᴇᴠᴀʟ ᴡɪᴛʜ ꜱᴜʙꜱᴇǫᴜᴇɴᴄᴇ ᴅʏɴᴀᴍɪᴄ ᴛɪᴍᴇ ᴡᴀʀᴘɪɴɢ

Given the above-mentioned definitions of sub-trajectories and foundation-model-driven similarity metrics, we instantiate an algorithm to find the $K$ most relevant sub-trajectories $\mathcal{T}_{\text{match}}$ from the offline dataset $\mathcal{D}_{\text{prior}}$ for each sub-trajectory $\boldsymbol{t}$ segmented from $\mathcal{D}_{\text{target}}$. Sub-trajectories may have variable lengths and temporal positioning within a trajectory caused by varying tasks, platforms, or demonstrators. We employ S-DTW to match the target sub-trajectories $\mathcal{T}_{\text{target}}$ to appropriate segment $\mathcal{T}_{\text{match}}$ in $\mathcal{D}_{\text{prior}}$ (Sec. 3.1). S-DTW scales naturally with these challenges and allows for retrieval from diverse, multi-task datasets. On deployment, subsequence dynamic time warping accepts a query sub-sequences from the target dataset, *i.e.*, $\boldsymbol{t}_{\text{target}}$, and uses dynamic programming to compute matches that are maximally aligned with the query $\mathcal{T}_{\text{match}} = \{\texttt{SDTW}(\boldsymbol{t}, \mathcal{D}_{\text{prior}}), \forall \boldsymbol{t} \in \mathcal{T}_{\text{target}}\}$ along with matching costs, $D$. To construct $\mathcal{D}_{\text{retrieval}}$, we select the $K$ matches with the lowest cost uniformly across the sub-trajectories in $\mathcal{T}_{\text{target}}$, *i.e.*, the same number of matches for each query until $K$ matches are retrieved. We note that the resulting set of matches can contain duplicates if the demonstrations share similar chunks, but argue that if a chunk occurs multiple times in the demonstrations, it is important to the task and should be *"up-weighted"* in the training set – in this case through duplicated retrieval. For each match, we also retrieve its corresponding language instruction. The training dataset then contains a union of the target dataset $\mathcal{D}_{\text{target}}$ and the retrieved dataset $\mathcal{D}_{\text{retrieval}}$, $\mathcal{D}_{\text{target}} \cup \mathcal{D}_{\text{retrieval}}$. This significantly larger, retrieval-augmented dataset can then be used to learn policies via imitation learning, leading to robust, generalizable policies.

## 4.5 Pᴜᴛᴛɪɴɢ ɪᴛ ᴀʟʟ ᴛᴏɢᴇᴛʜᴇʀ: STRAP

To start the retrieval process, we encode image observations in $\mathcal{D}_{\text{target}}$ and $\mathcal{D}_{\text{prior}}$ using a vision foundation model, *e.g.*, DINOv2 (Oquab et al., 2023) or CLIP (Radford et al., 2021). To best leverage the multi-task trajectories in $\mathcal{D}_{\text{prior}}$, we split the demonstrations in $\mathcal{D}_{\text{target}}$ into atomic chunks based on a low-level motion heuristic. Then we generate matches between chunked $\mathcal{D}_{\text{target}}$ and $\mathcal{D}_{\text{prior}}$ and construct $\mathcal{D}_{\text{retrieval}}$ by selecting the top $K$ matches uniformly across all chunks. Combining $\mathcal{D}_{\text{retrieval}}$

Stove-Moka    Bowl-Drawer    Soup-Cheese    Mug-Mug    Book-Caddy          Table          Sink          Stove

Figure 12: **Tasks in $\mathcal{D}_{\text{target}}$:** LIBERO-10 (left) and real-world DROID-Kitchen (right). For $\mathcal{D}_{\text{prior}}$ please refer to LIBERO-90 (Liu et al., 2024) for the simulated and Fig. 28. for real-world tasks.

with $\mathcal{D}_{\text{target}}$ forms our dataset for learning a policy. In a standard policy learning setting, noisy retrieval data can lead to negative transfer, *e.g.*, when observations similar to the target data are labeled with actions that achieve a different task. Without conditioning, such contaminated samples hurt the policy's downstream performance. We propose to use a language-conditioned policy to deal with this inconsistency. With conditioning, the policy can distinguish between samples from different tasks, separating misleading from expert actions while benefiting from positive transfer from the additional training data and context of the language conditioning.

We use behavior cloning (BC) to learn a visuomotor policy $\pi$ similar to Haldar et al. (2024); Nasiriany et al. (2024). We choose a transformer-based (Vaswani, 2017) architecture feeding in a history of the last $h$ observations $s_{t-h:t}$ and predicting a chunk of $h$ future actions using a Gaussian mixture model action head. We sample batches from the union of $\mathcal{D}_{\text{target}}$ and $\mathcal{D}_{\text{retrieval}}$, as in $\mathcal{B} \sim \mathcal{D}_{\text{target}} \cup \mathcal{D}_{\text{retrieval}}$. As proposed in Haldar et al. (2024) we compute the multi-step action loss and add an L2 regularization term over the model weights $\theta$, resulting in the following loss function:

$$\mathcal{L}(\mathcal{B};\theta) = \frac{1}{|\mathcal{B}|} \sum_{(s_{i-h:i},a_{i:i+h},l)\in\mathcal{B}} -\log(\pi_\theta(a_{i:i+h}|s_{i-h:i},l)) + \lambda\|\theta\|_2^2 \tag{3}$$

with policy $\pi_\theta$ and hyperparameter $\lambda$ controlling the regularization.

## 5 EXPERIMENTS AND RESULTS

### 5.1 EXPERIMENTAL SETUP

**Task Definition:** We demonstrate the efficacy of STRAP in simulation on the LIBERO benchmark (Liu et al., 2024), in two real-world scenarios following the DROID (Khazatsky et al., 2024) hardware setup. Fig. 12 shows the target tasks and samples from the retrieval datasets. For more task details please refer to Appendix A.2.1.

- **LIBERO:** We evaluate STRAP on 10 long-horizon tasks of the LIBERO benchmark (Liu et al., 2024) which include diverse objects, layouts, and backgrounds. We randomly sample 5 demonstrations from LIBERO-10 as $\mathcal{D}_{\text{target}}$ and utilize the 4500 trajectories in LIBERO-90 as $\mathcal{D}_{\text{prior}}$. The evaluation environments randomize the target object poses, providing an ideal test bed for robustness. We report the top five tasks and average LIBERO-10 performance in Tab. 1 and provide the remaining ones in the appendix (Tab. 3).
- **DROID-Kitchen:** Scaling STRAP to more realistic scenarios, we evaluate STRAP on vegetable pick-and-place in three real kitchen environments. We collect 150 multi-task demonstrations across scenes in $\mathcal{D}_{\text{prior}}$ (Kitchen), with 50 per scene. Each demo includes two unique tasks, randomly sampled from three options, with varied object poses and appearances. $\mathcal{D}_{\text{target}}$ contains three demos of a single task per environment. To assess STRAP 's scalability, we sample 5000 demos from the DROID dataset and 50 from $\mathcal{D}_{\text{target}}$ 's environment (Kitchen-DROID). During evaluation, object poses vary within a $20 \times 20cm$ grid. Success rates are measured as partial (pick, 0.5) and full (pick and place, 1.0), normalized to a $0 - 100$ scale.

**Baselines and Ablations:** We compare STRAP to the following baselines and ablations and refer the reader to Appendix A.1 for implementation details and Appendix A.3 for extensive ablations.

- **Behavior Cloning** (BC) behavior cloning using a transformer-based policy trained on $\mathcal{D}_{\text{target}}$;
- **Fine-tuning** (FT) behavior cloning using a transformer-based policy pre-trained on $\mathcal{D}_{\text{prior}}$ and fine-tuned on $\mathcal{D}_{\text{target}}$;
- **Multi-task Policy** (MT) transformer-based multi-task policy trained on $\mathcal{D}_{\text{prior}}$ and $\mathcal{D}_{\text{target}}$;

Table 1: **Baselines:** Performance of baselines, ablations and variations of STRAP on the LIBERO-10 tasks (Fig. 12). DINOv2 and CLIP features perform similarly, making STRAP flexible in the encoder choice. Additional results in Tab. 3. **Bold** indicates best and underline runner-up results.

| Task | Stove-Moka | Bowl-Cabinet | Soup-Cheese | Mug-Mug | Book-Caddy | LIBERO-10 |
|---|---|---|---|---|---|---|
| BC | $77.3\% \pm 4.4$ | $71.3\% \pm 5.7$ | $27.3\% \pm 2.2$ | $38.0\% \pm 5.7$ | $75.3\% \pm 1.4$ | $37.9\% \pm 27.2$ |
| FT | $\underline{86.0\%} \pm 1.4$ | $91.0\% \pm 0.7$ | $38.0\% \pm 2.8$ | $43.0\% \pm 0.7$ | $\mathbf{100.0\%} \pm \mathbf{0.0}$ | $\underline{51.7\%} \pm 35.7$ |
| MT | $66.0\% \pm 14.1$ | $45.0\% \pm 30.4$ | $19.0\% \pm 13.4$ | $31.0\% \pm 16.3$ | $\mathbf{100.0\%} \pm \mathbf{0.0}$ | $37.7\% \pm 32.6$ |
| BR | $80.0\% \pm 1.6$ | $72.0\% \pm 7.7$ | $26.0\% \pm 5.3$ | $40.0\% \pm 8.6$ | $16.0\% \pm 1.9$ | $33.4\% \pm 25.1$ |
| FR | $76.0\% \pm 6.6$ | $54.7\% \pm 12.0$ | $24.7\% \pm 8.6$ | $29.3\% \pm 1.4$ | $52.0\% \pm 5.9$ | $33.1\% \pm 23.0$ |
| D-S | $70.7\% \pm 7.9$ | $65.3\% \pm 2.0$ | $18.0\% \pm 3.4$ | $16.0\% \pm 0.9$ | $57.3\% \pm 2.9$ | $28.4\% \pm 26.4$ |
| D-T | $78.7\% \pm 2.7$ | $75.3\% \pm 2.7$ | $37.3\% \pm 6.6$ | $\underline{63.3\%} \pm 3.6$ | $79.0\% \pm 5.0$ | $41.4\% \pm 30.8$ |
| STRAP (CLIP, $K{=}100$) | $\underline{86.0\%} \pm 4.1$ | $90.7\% \pm 2.2$ | $\underline{42.0\%} \pm 0.9$ | $54.7\% \pm 3.3$ | $83.3\% \pm 3.0$ | $44.9\% \pm 32.7$ |
| STRAP (DINOv2, $K{=}100$) | $85.3\% \pm 2.2$ | $\underline{91.3\%} \pm 2.2$ | $\mathbf{42.7\%} \pm \mathbf{7.2}$ | $57.3\% \pm 7.7$ | $85.3\% \pm 2.8$ | $45.6\% \pm 32.6$ |
| STRAP (DINOv2, best $K$) | $\mathbf{94.0\%} \pm \mathbf{1.4}$ | $\mathbf{96.0\%} \pm \mathbf{0.0}$ | $\mathbf{42.7\%} \pm \mathbf{7.2}$ | $\mathbf{69.0\%} \pm \mathbf{0.7}$ | $\underline{94.0\%} \pm 4.2$ | $\mathbf{58.1\%} \pm \mathbf{32.0}$ |

- **BR** (BehaviorRetrieval) (Du et al., 2023) prior work that trains a VAE on state-action pairs for retrieval and uses cosine similarity to retrieve single state-action pairs;
- **FR** (FlowRetrieval) (Lin et al., 2024) same setup as BR but VAE is trained on pre-computed optical flow from GMFlow (Xu et al., 2022);
- **D-S** (DINOv2 state) same as BR and FR but uses off-the-shelf DINOv2 (Oquab et al., 2023) features instead of training a VAE;
- **D-T** (DINOv2 trajectory) retrieves *full* trajectories (rather than sub-trajectories) with S-DTW and DINOv2 features;

## 5.2 Experimental Evaluation

Our evaluation aims to address the following questions: (1) Does *sub-trajectory retrieval* improve performance in few-shot imitation learning? (2) How effective are the representations from *vision-foundation models* for retrieval? (3) What types of matches are identified by *S-DTW*?

**Does *sub-trajectory retrieval* improve performance in few-shot imitation learning?** STRAP outperforms the retrieval baselines BR and FR on average by $+24.7\%$ and $+25.0\%$ across all 10 tasks (Tab. 1). These results demonstrate the policy's robustness to varying object poses. BC represents a strong baseline on the LIBERO task as the benchmark's difficulty comes from pose variations during evaluation. By memorizing the demonstrations, BC achieves high success rates, outperforming BR and FR by $+4.5\%$ and $+4.8\%$ across all 10 tasks. In our real-world experiments, BC performs much worse due to the increased randomization during evaluation. The policy replays the demonstrations in $\mathcal{D}_{\text{target}}$, failing to adapt to new object poses.

Pre-training a policy on $\mathcal{D}_{\text{prior}}$ and fine-tuning it on $\mathcal{D}_{\text{target}}$ (FT) emerges as the most competitive baseline underperforming STRAP by only $-6.4\%$. Training a multi-task policy (MT) matches BC on average but shows improvements when $\mathcal{D}_{\text{prior}}$ contains demonstrations or environments overlapping with $\mathcal{D}_{\text{target}}$. Introducing larger randomization to $\mathcal{D}_{\text{prior}}$ and

Table 2: **Real-world results:** DROID-Kitchen

| | Kitchen | | | Kitchen+DROID | | |
|---|---|---|---|---|---|---|
| | Table | Sink | Stove | Table | Sink | Stove |
| BC | 12.50 | 10.00 | 14.28 | 12.50 | $\underline{10.00}$ | 14.28 |
| FT | $\underline{20.00}$ | 27.27 | 30.43 | $\underline{28.00}$ | 8.69 | $\underline{22.72}$ |
| MT | 4.34 | $\underline{31.57}$ | $\underline{45.00}$ | 2.00 | 0.00 | 0.00 |
| STRAP | **36.36** | **61.36** | **57.12** | **56.81** | **63.04** | **45.45** |

the evaluation as part of the real-world experiments hurts the performance of both methods. Specifically, we find checkpoint selection for FT difficult as it's easy for the policy to under- or overfit the demonstrations, leading to degenerate behavior or exact replay. We found MT training challenging as the policy (Haldar et al., 2024; Nasiriany et al., 2024) sometimes ignores the language instruction and solves a task seen in $\mathcal{D}_{\text{prior}}$ instead of the conditioned $\mathcal{D}_{\text{target}}$. We hypothesize that this behavior emerges due to the data imbalance of $\mathcal{D}_{\text{prior}}$ and $\mathcal{D}_{\text{target}}$. Finally, augmenting $\mathcal{D}_{\text{prior}}$ with 5000 trajectories from the DROID dataset amplifies these challenges leading to an even larger performance gap. In our real-world evaluations, we find STRAP to experience surprising generalization behavior to poses unseen in $\mathcal{D}_{\text{target}}$. The policy further shows recovery behavior, completing the task even when the initial grasp fails and alters the object's pose. Since STRAP's policy training stage is independent of the size of $\mathcal{D}_{\text{prior}}$ but the dataset size is determined by hyperparameter $K$, it can naturally deal with adding in larger datasets like DROID maintaining performance.

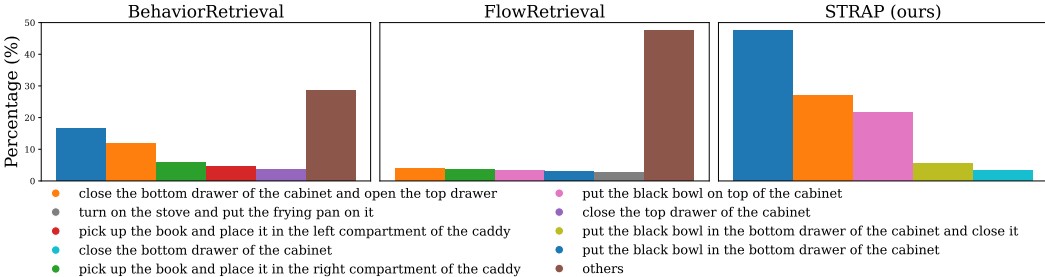

Figure 13: **Tasks distribution** in $\mathcal{D}_{\text{retrieval}}$ for different retrieval methods with target task *"put the black bowl in the bottom drawer of the cabinet and close it"*.

**How important are *sub-trajectories* for retrieval and how many should be retrieved?** To investigate the efficacy of sub-trajectories, we compare sub-trajectory retrieval with S-DTW (STRAP) to retrieving full trajectories with S-DTW (D-T) in Tab. 1. We find sub-trajectory retrieval to improve performance by $+4.1\%$ across all 10 tasks. We hypothesize that full trajectories can contain segments irrelevant to the task, effectively hurting performance and reducing the accuracy of the cumulative cost. Varying the number of retrieved segments $K$, we find that the optimal value for $K$ is highly task-dependent with some tasks benefiting from retrieving less and some from retrieving more data. We hypothesize that $K$ depends on whether tasks leverage (positive transfer) or suffer (negative transfer) from multi-task training. We report the results for the best $K$ in Tab. 1 and provide the full search in Tab. 9.

**How effective are the representations from *vision-foundation models* for retrieval?** Next, we ablate the choice of foundation model representation in STRAP with fixed $K = 100$. We compare CLIP, a model trained through supervised learning on image-text pairs, with DINOv2, a self-supervised model trained on unlabeled images. We don't find any representation to significantly outperform the other with DINOv2 separated from CLIP by only $+0.7\%$ across all 10 tasks. To show the efficacy of vision-foundation models for retrieval, we replace the in-domain feature extractors from prior work (BR, FR) trained on $\mathcal{D}_{\text{prior}}$ with an off-the-shelf DINOv2 encoder model (D-S). Comparing them in their natural configuration, *i.e.*, state-based retrieval using cosine similarity allows for a side-by-side comparison of the representations. Tab. 1 shows the choice of representation to depend on the task with no method outperforming the others on all tasks. Since D-S has no notion of dynamics and task semantics due to single-state retrieval, BR and FR outperform it by $+5.0\%$ and $+4.7\%$, respectively. We highlight that vision foundation models don't have to be trained on $\mathcal{D}_{\text{prior}}$ and scale much better with increasing amounts of trajectory data and on unseen tasks.

**What types of matches are identified by *S-DTW*?** To understand what data STRAP retrieves, we visualize the distribution over tasks as a function of $\mathcal{D}_{\text{retrieval}}$ proportion in Figure 13. The figure visualizes the top five tasks retrieved and accumulates the rest into the "others" category. It becomes clear that STRAP retrieves semantically relevant data – each task shares at least one sub-task with the target task. For example, *"put the black bowl in the bottom drawer of the cabinet"*, *"close the bottom drawer of the cabinet ..."* (Fig. 3). Furthermore, STRAP's retrieval is sparse, only selecting data from 5/90 semantically relevant tasks and ignoring irrelevant ones. We observe that DINOv2 features are surprisingly agnostic to different environment textures, retrieving data from the same task but in a different environment (*c.f.* Fig. 13, *"put the black bowl in the bottom drawer of the cabinet and close it"*). Furthermore, DINOv2 is robust to object poses retrieving sub-trajectories that "close the drawer" with the bowl either on the table or in the drawer (*c.f.* Fig. 31, *"close the bottom drawer of the cabinet and open the top drawer"*). Trained on optical flow, FR has no notion of visual appearance, failing to retrieve most of the semantically relevant data.

## 6 CONCLUSION

We introduce STRAP as an innovative approach for leveraging visual foundation models in few-shot robotics manipulation, eliminating the need to train on the entire retrieval dataset, and allowing it to scale with minimal compute overhead. By focusing on sub-trajectory retrieval using S-DTW, STRAP improves data utilization and captures dynamics more effectively. Overall, our approach outperforms standard fine-tuning and multi-task approaches as well as state-of-the-art retrieval methods by a large margin, showcasing robust performance in challenging real-world scenarios.

**Acknowledgements:** The authors thank the anonymous reviewers for their valuable feedback, Soroush Nasiriany for answering our many questions on the robocasa and robosuite codebase, Li-Heng Lin and Yuchen Cui for helping us understand policy learning with retrieved data, colleagues and fellow interns at Bosch for the fruitful discussions. The work has received funding from Amazon (CC OTH 00203375 2024 TR), ARL (W911NF2420191), and Robert Bosch LLC (most of this work was done during Marius Memmel's 2024 summer internship project).

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

# A   APPENDIX

## A.1   SIMULATION EXPERIMENTS

Table 3: **Baselines (sim):** Performance of different methods on LIBERO-10 tasks in simulation. **Bold** indicates best and underline runner-up results.

| Method | Mug-Microwave | Moka-Moka | Soup-Sauce | Cream-Cheese-Butter | Mug-Pudding |
|---|---|---|---|---|---|
| BC | $28.0\% \pm 0.9$ | $0.0\% \pm 0.0$ | $17.3\% \pm 4.5$ | $26.7\% \pm 4.3$ | $18.0\% \pm 2.5$ |
| Fine-Tuning | $\mathbf{38.0\% \pm 5.7}$ | $0.0\% \pm 0.0$ | $5.0\% \pm 2.1$ | $\mathbf{81.0\% \pm 3.5}$ | $\mathbf{35.0\% \pm 3.5}$ |
| Multi-Task | $10.0\% \pm 7.1$ | $0.0\% \pm 0.0$ | $\underline{24.0\% \pm 17.0}$ | $\underline{73.0\% \pm 13.4}$ | $9.0\% \pm 2.1$ |
| BR (Du et al., 2023) | $28.7\% \pm 3.9$ | $0.0\% \pm 0.0$ | $13.3\% \pm 3.8$ | $32.0\% \pm 4.3$ | $26.0\% \pm 1.9$ |
| FR (Lin et al., 2024) | $27.3\% \pm 1.4$ | $0.0\% \pm 0.0$ | $11.3\% \pm 3.0$ | $41.3\% \pm 5.5$ | $14.7\% \pm 1.1$ |
| D-S | $30.0\% \pm 3.4$ | $0.0\% \pm 0.0$ | $4.7\% \pm 0.5$ | $16.0\% \pm 5.7$ | $6.0\% \pm 0.9$ |
| D-T | $\underline{34.7\% \pm 2.0}$ | $0.0\% \pm 0.0$ | $4.7\% \pm 1.1$ | $27.3\% \pm 4.5$ | $14.0\% \pm 3.4$ |
| STRAP (CLIP, $K$=100) | $30.0\% \pm 2.5$ | $0.0\% \pm 0.0$ | $8.7\% \pm 6.3$ | $29.3\% \pm 10.5$ | $24.0\% \pm 4.3$ |
| STRAP (DINOv2, $K$=100) | $29.3\% \pm 2.7$ | $0.0\% \pm 0.0$ | $16.7\% \pm 2.0$ | $29.3\% \pm 11.3$ | $18.7\% \pm 1.4$ |
| STRAP (DINOv2, best $K$) | $32.0\% \pm 5.7$ | $0.0\% \pm 0.0$ | $\mathbf{61.0\% \pm 6.4}$ | $61.0\% \pm 0.7$ | $\underline{31.0\% \pm 2.1}$ |

**Hyperparameters:**   All results are reported over 3 training and evaluation seeds (1234, 42, 4325). We fixed both the number of segments retrieved to 100, the camera viewpoint to the agent view image for retrieval, and the number of expert demonstrations to 5. We use the agent view (exocentric) observations for the retrieval and train policies on both agent view and in-hand observations. Our transformer policy was trained for 300 epochs with batch size 32 and an epoch every 200 gradient steps.

**Baseline implementation details:**   Following Lin et al. (2024), we retrieve single-state action pairs for the state-based retrieval baselines (BR, FR, D-S) and pad them by also retrieving the states from $t - h$ to $t + h - 1$ to make the samples compatible with our transformer-based policy. We refer the reader to Appendix A.3 for extensive ablation.

**Remaining results on LIBERO-10**   Tab. 3 shows the results for the remaining LIBERO-10 task not reported in the main sections. Both FR and BR outperform STRAP on the Cream-Cheese-Butter task. We hypothesize that our chunking heuristic generates sub-optimal sub-trajectories (too long) causing them to contain multiple different semantic tasks, leading to worse matches in our retrieval datasets and eventually in decreasing downstream performance.

## A.2   REAL-WORLD EXPERIMENTS

**Setup & hyperparameters:**   For retrieval, we average the embeddings per time-step across the left, right, and in-hand camera observations and fix the number of segments retrieved to 100. We train policies on all three image observations for 200 epochs with batch size 32 and an epoch every 100 gradient steps. We initialize the ResNet-18 (He et al., 2015) vision encoders of our policy with weights pre-trained on ImageNet (Deng et al., 2009). We use the DROID-setup (Khazatsky et al., 2024) – running at 15Hz – to collect demonstrations and deploy our policies on the real robot.

### A.2.1   FRANKA-PEN-IN-CUP

**Task:** We evaluate STRAP's ability to retrieve from "unrelated" tasks in a pen-in-cup scenario. The offline dataset $\mathcal{D}_{\mathrm{prior}}$ consists of 100 single-task demonstrations across 10 tasks and $\mathcal{D}_{\mathrm{target}}$ contains 3 demonstrations for the pen-in-cup task. While some share sub-tasks, *e.g.*, *"put the pen next to the markers"*, *"put the screwdriver into the cup"*, *"put the green marker into the mug"*, others are unrelated, *e.g.*, *"make a hotdog"*, *"push over the box"*, *"pull the cable to the right"*. Note that the target task is not part of $\mathcal{D}_{\mathrm{prior}}$! We demonstrate STRAP's ability to an unseen pose, and object and target appearance.

Table 4: **Franka-Pen-in-Cup:** real-world results.

| Pen-in-Cup | *base* | | *OOD* | |
|---|---|---|---|---|
| | Pick | Place | Pick | Place |
| BC | 100% | 100% | 0% | 0% |
| STRAP | 100% | 90% | **100%** | **100%** |

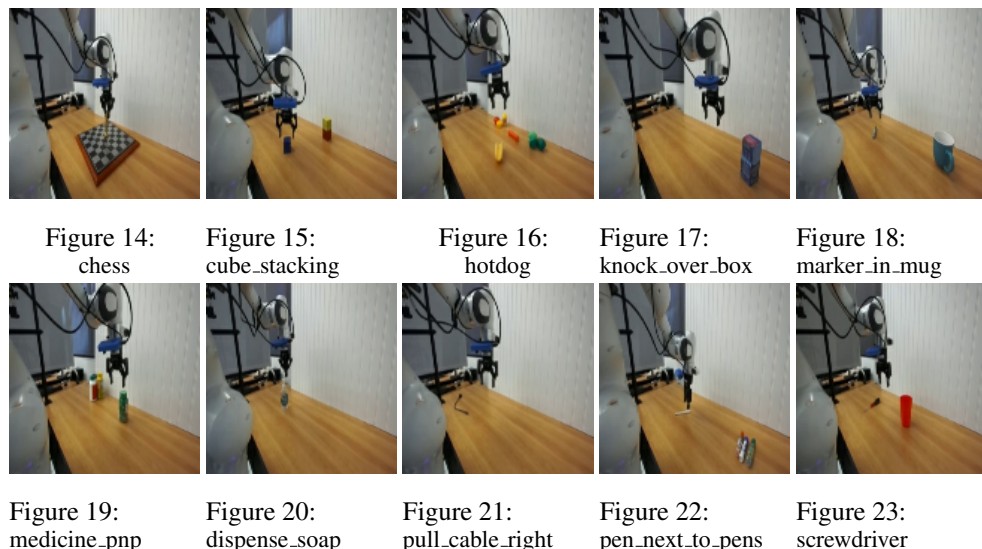

Figure 14: chess

Figure 15: cube_stacking

Figure 16: hotdog

Figure 17: knock_over_box

Figure 18: marker_in_mug

Figure 19: medicine_pnp

Figure 20: dispense_soap

Figure 21: pull_cable_right

Figure 22: pen_next_to_pens

Figure 23: screwdriver

Figure 24: **Franka-Pen-in-Cup:** Environment setup for the tasks in $\mathcal{D}_{\text{prior}}$.

| | Language Instructions $\mathcal{D}_{\text{prior}}$ |
|---|---|
| chess | "Move the king to the top right of the chess board" |
| cube_stacking | "Stack the blue cube on top of the tower" |
| hotdog | "Put the hotdog in the bun" |
| knock_over_box | "Knock over the box" |
| marker_in_mug | "Put the marker in the mug" |
| medicine_pnp | "Pick up the medicine box on the right and put it next to the other medicine boxes" |
| dispense_soap | "Press the soap dispenser" |
| pull_cable_right | "Pull the cable to the right" |
| pen_next_to_pens | "Put the pen next to the markers" |
| screwdriver | "Pick up the screwdriver and put it in the cup" |

Table 5: **Franka-Pen-in-Cup:** Task/language instructions for the real-world dataset $\mathcal{D}_{\text{prior}}$.

### A.2.2 DROID-KITCHEN

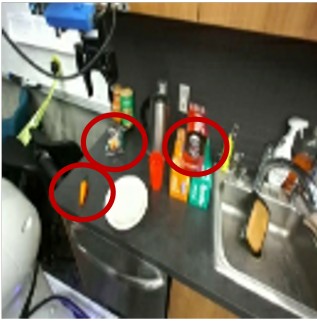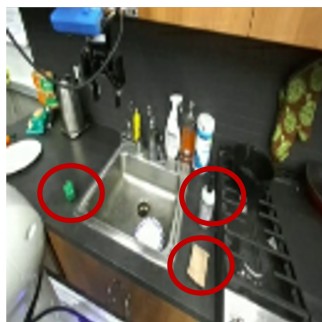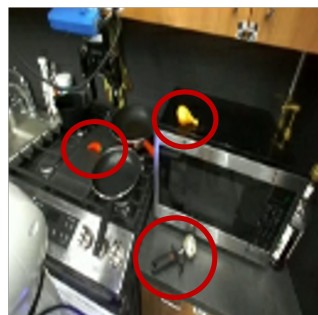

Figure 25:
table

Figure 26:
sink

Figure 27:
stove

Figure 28: **DROID-Kitchen:** Environment setup for the tasks in $\mathcal{D}_{\text{prior}}$. Task-relevant objects are marked by red circles. From left to right, the objects are **table:** carrot, chips bag, can; **sink:** pepper, soap dispenser, sponge; **stove:** chili, chicken, utensil. We randomize the object pose and type, *e.g.*, color or style during data collection.

| | Language Instructions $\mathcal{D}_{\text{prior}}$ | Language Instruction $\mathcal{D}_{\text{target}}$ |
|---|---|---|
| table | "press the soap dispenser", "pick up the sponge and put it in the sink", "pick up the pepper and put it in the sink" | "pick up the pepper and put it in the sink" |
| sink | "pick up the carrot and put it on the plate", "pick up the chips bag and put it on the plate", "pick up the can and move it next to the table" | "pick up the can and move it next to the table" |
| stove | "pick up the chicken wing and put it in the pan", "pick up the utensil and put it in the pan", "pick up the chili and put it in the pan" | "pick up the chili and put it in the pan" |

Table 6: **DROID-Kitchen:** Task/language instructions for the real-world datasets $\mathcal{D}_{\text{prior}}$ and $\mathcal{D}_{\text{target}}$. Each task in $\mathcal{D}_{\text{prior}}$ consists of two unique tasks randomly sampled from the available instructions. The chosen instructions are concatenated by the filler ", then". In the table environment, for example, a task might be *"pick up the sponge and put it in the sink, then press the soap dispenser"*.

### A.3 ABLATIONS

Table 7: **Ablations - Retrieval Method:** We explore different approaches for trajectory-based retrieval. Besides the heuristic reported in the main paper, we experiment with a sliding window approach that segments a trajectory into sub-trajectories of equal length (here: 30). We use S-DTW for both sliding window sub-trajectories and full trajectories.

| Method | Stove-Moka | Bowl-Cabenet | Mug-Microwave | Moka-Moka | Soup-Cream-Cheese |
|---|---|---|---|---|---|
| Sub-traj | $76.0\% \pm 4.71$ | $\mathbf{75.33\% \pm 2.72}$ | $26.0\% \pm 1.89$ | $0.0\% \pm 0.00$ | $\mathbf{37.33\% \pm 6.62}$ |
| Full traj | $\mathbf{78.67\% \pm 2.72}$ | $68.67\% \pm 1.44$ | $\mathbf{34.67\% \pm 1.96}$ | $0.0\% \pm 0.00$ | $28.67\% \pm 3.81$ |

| Method | Soup-Sauce | Cream-Cheese-Butter | Mug-Mug | Mug-Pudding | Book-Caddy |
|---|---|---|---|---|---|
| Sub-traj | $\mathbf{40.00\% \pm 0.94}$ | $\mathbf{27.33\% \pm 2.18}$ | $\mathbf{63.33\% \pm 3.57}$ | $\mathbf{30.00\% \pm 3.40}$ | $\mathbf{79.0\% \pm 4.95}$ |
| Full traj | $4.67\% \pm 1.09$ | $\mathbf{27.33\% \pm 4.46}$ | $43.33\% \pm 1.09$ | $14.0\% \pm 3.4$ | $68.0\% \pm 5.66$ |

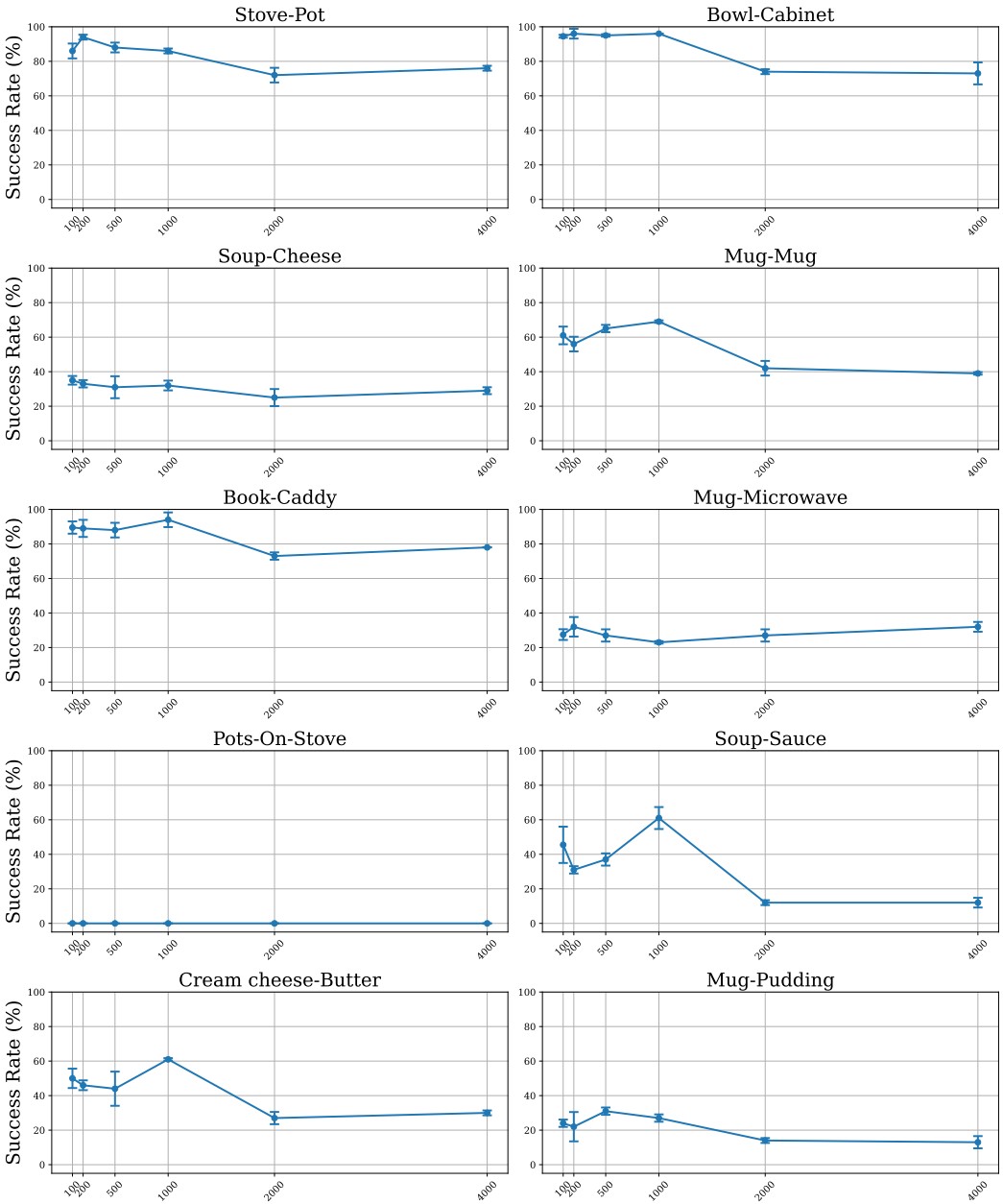

Figure 29: **Ablations - $K$ (Num Segments Retrieved):** The figure shows mean success and standard deviation for different values of $K$.

Table 8: **Ablations -** $K$ **(Num Segments Retrieved):** Tuning $K$ improves the success rates reported in Tab. 1 on 8/10 tasks. The optimal value for $K$ is highly task-dependent with some tasks benefiting from retrieving less (*Stove-Pot, Bowl-Cabinet, Soup-Cheese, Mug-Microwave*) and some from retrieving more (*Mug-Mug, Book-Caddy, Soup-Sauce, Cream cheese-Butter, Mug-Pudding*) data. We hypothesize that $K$ depends on whether tasks leverage (positive transfer) or suffer (negative transfer) from multi-task training.

| K | Stove-Moka | Bowl-Cabinet | Soup-Cheese | Mug-Mug | Book-Caddy |
|---|---|---|---|---|---|
| $K = 100$ | $86.0\% \pm 4.3$ | $94.5\% \pm 0.83$ | $\mathbf{35.0\% \pm 2.5}$ | $61.0\% \pm 5.12$ | $89.5\% \pm 3.56$ |
| $K = 200$ | $\mathbf{94.0\% \pm 1.41}$ | $\mathbf{96.0\% \pm 2.83}$ | $33.0\% \pm 2.12$ | $56.0\% \pm 4.24$ | $89.0\% \pm 4.95$ |
| $K = 500$ | $88.0\% \pm 2.83$ | $95.0\% \pm 0.71$ | $31.0\% \pm 6.36$ | $65.0\% \pm 2.12$ | $88.0\% \pm 4.24$ |
| $K = 1000$ | $86.0\% \pm 1.41$ | $96.0\% \pm 0.00$ | $32.0\% \pm 2.83$ | $\mathbf{69.0\% \pm 0.71}$ | $\mathbf{94.0\% \pm 4.24}$ |
| $K = 2000$ | $72.0\% \pm 4.24$ | $74.0\% \pm 1.41$ | $25.0\% \pm 4.95$ | $42.0\% \pm 4.24$ | $73.0\% \pm 2.12$ |
| $K = 4000$ | $76.0\% \pm 1.41$ | $73.0\% \pm 6.36$ | $29.0\% \pm 2.12$ | $39.0\% \pm 0.71$ | $78.0\% \pm 0.0$ |

| K | Mug-Microwave | Pots-On-Stove | Soup-Sauce | Cream cheese-Butter | Mug-Pudding |
|---|---|---|---|---|---|
| $K = 100$ | $27.5\% \pm 3.11$ | $0.0\% \pm 0.00$ | $45.5\% \pm 10.52$ | $50.0\% \pm 5.61$ | $24.0\% \pm 2.12$ |
| $K = 200$ | $\mathbf{32.0\% \pm 5.66}$ | $0.0\% \pm 0.00$ | $31.0\% \pm 2.12$ | $46.0\% \pm 2.83$ | $22.0\% \pm 8.49$ |
| $K = 500$ | $27.0\% \pm 3.54$ | $0.0\% \pm 0.00$ | $37.0\% \pm 3.54$ | $44.0\% \pm 9.9$ | $\mathbf{31.0\% \pm 2.12}$ |
| $K = 1000$ | $23.0\% \pm 0.71$ | $0.0\% \pm 0.00$ | $\mathbf{61.0\% \pm 6.36}$ | $\mathbf{61.0\% \pm 0.71}$ | $27.0\% \pm 2.12$ |
| $K = 2000$ | $27.0\% \pm 3.54$ | $0.0\% \pm 0.0$ | $12.0\% \pm 1.41$ | $27.0\% \pm 3.54$ | $14.0\% \pm 1.41$ |
| $K = 4000$ | $32.0\% \pm 2.83$ | $0.0\% \pm 0.0$ | $12.0\% \pm 2.83$ | $30.0\% \pm 1.41$ | $13.0\% \pm 3.54$ |

Table 9: **Ablations - Retrieval Seeds:** We run `STRAP` on different retrieval seeds on a subset of LIBERO-10 tasks using 10 expert demos as $\mathcal{D}_{\text{retrieval}}$. We report results over all possible combinations of 3 training and 3 retrieval seeds

| Method | Stove-Moka | Mug-Cabinet | Book-Caddy |
|---|---|---|---|
| BC Baseline | $81.78\% \pm 2.6$ | $83.11\% \pm 2.69$ | $93.11\% \pm 1.57$ |
| STRAP | $\mathbf{86.89\% \pm 1.51}$ | $\mathbf{88.67\% \pm 2.11}$ | $\mathbf{98.0\% \pm 1.04}$ |

## A.4 ADDITIONAL BASELINES

Table 10: **Diffusion Policy Baselines:** Performance on LIBERO-10 tasks using diffusion policies (DP) (Chi et al., 2023) without language conditioning for BehaviorRetrieval (BR) (Du et al., 2023), FlowRetrieval (FR) (Lin et al., 2024). These experiments replicate the training setup for BR and FR.

| Method | Stove-Moka | Bowl-Cabinet | Soup-Cheese | Mug-Mug | Book-Caddy |
|---|---|---|---|---|---|
| BR | $36.67\% \pm 1.44$ | $68.0\% \pm 2.49$ | $34.0\% \pm 2.49$ | $55.33\% \pm 1.44$ | $42.0\% \pm 1.63$ |
| FR | $68.67\% \pm 2.37$ | $56.0\% \pm 4.32$ | $18.0\% \pm 3.4$ | $56.0\% \pm 3.4$ | $35.33\% \pm 6.28$ |

| Method | Mug-Microwave | Pots-On-Stove | Soup-Sauce | Cream cheese-Butter | Mug-Pudding |
|---|---|---|---|---|---|
| BR | $30.67\% \pm 0.54$ | $0.00\% \pm 0.00$ | $10.67\% \pm 1.96$ | $24.0\% \pm 0.94$ | $9.33\% \pm 1.44$ |
| FR | $32.67\% \pm 3.31$ | $68.0\% \pm 2.49$ | $6.0\% \pm 0.00$ | $35.33\% \pm 0.54$ | $8.0\% \pm 1.89$ |

## A.5 COMPLEXITY AND SCALABILITY

### A.5.1 ENCODING TRAJECTORIES WITH VISION FOUNDATION MODELS

The embeddings for $\mathcal{D}_{prior}$ can be precomputed and reused for every new retrieval process. Embedding a dataset with total timesteps $T$ and number of camera views $V$ scales linear with $\mathcal{O}(T * V)$. We benchmark Huggingface's DINOv2 implementation[3] on an NVIDIA L40S 46GB using batch size 32. Encoding a single image takes $2.83ms \pm 0.08$ (average across 25 trials). The wall clock time for encoding the entire DROID dataset ( 18.9M timesteps, single-view) therefore sums up to only  26h.

---

[3] https://huggingface.co/docs/transformers/en/model_doc/dinov2

### A.5.2 SUBSEQUENCE DYNAMIC TIME WARPING

In contrast to the embedding process, retrieval must be run for every deployment scenario. S-DTW consists of two stages: computing the distance matrix $D$ and finding the shortest path via dynamic programming. Computing the distance matrix has complexity $\mathcal{O}(n \cdot m \cdot E)$ with $E$ the embedding dimension, $n$ the length of the sub-trajectory in $\mathcal{D}_{\text{target}}$, and $m$ the length of the trajectory in $\mathcal{D}_{\text{prior}}$. DTW and backtracking have complexities of $\mathcal{O}(n \cdot m)$ and $\mathcal{O}(n)$, respectively. These stages have to be run sequentially for each sub-trajectory ($\in \mathcal{D}_{target}$) and trajectory ($\in \mathcal{D}_{prior}$) but don't depend on the other (sub-)trajectories. Therefore, STRAP has a runtime complexity of $\mathcal{O}(N * M)$ with N the number of sub-trajectories in $\mathcal{D}_{target}$ and M the number of trajectories in $\mathcal{D}_{prior}$. Our implementation largely follows[4]. We use numba[5] to compile python functions into optimized machine code and warm-start every method by running it three times. Following the statistics of DROID, we choose a trajectory length of 250 ($\mathcal{D}_{target}$ and $\mathcal{D}_{prior}$) and a single demonstration from $\mathcal{D}_{target}$ split into 5 sub-trajectories of length 50 each and embed each timestep into a 768-dimensional vector mimicking DINOv2 embeddings. We benchmark S-DTW and report the wall clock time (average over 10 trials) in Fig. 30. For an offline dataset the size of DROID (76k), retrieval takes approximately $300sec$. Note that computing the distance matrix can be expressed as matrix multiplications and can leverage GPU deployment and custom CUDA kernels for even greater speedup.

### A.5.3 POLICY TRAINING

The training process also has to be repeated for every deployment scenario. We use robomimic[6] and train policies for 200 epochs with a batch size of 32 and 100 gradient steps per epoch on an NVIDIA L40S 46GB. Training a single policy takes $35min \pm 4$ (average over 10 trials).

### A.5.4 INCREASING THE DATASET SIZE

Overall, STRAP scales linearly with new trajectories added to $\mathcal{D}_{prior}$. STRAP encodes the new trajectories using an off-the-shelf vision foundation model, eliminating the need to re-train the encoder like in previous approaches (BR, FR). Retrieving data with S-DTW scales linearly with the size of $\mathcal{D}_{prior}$, allowing for retrieval within 5min even from the largest available datasets like DROID. Finally, STRAP 's policy learning stage is independent of the size of $\mathcal{D}_{prior}$ and only depends on the amount of retrieved data $K$, making it more scalable than common pre-training + fine-tuning or multi-task approaches that have to be re-trained when new trajectories are added to $\mathcal{D}_{prior}$.

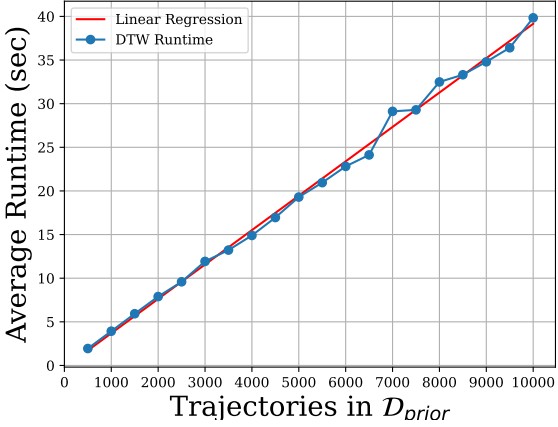

Figure 30: **STRAP Runtime:** We benchmark STRAP's retrieval step on varying sizes of $\mathcal{D}_{\text{prior}}$ and report the average wall clock time over 10 trials.

---

[4]https://www.audiolabs-erlangen.de/resources/MIR/FMP/C7/C7S2_SubsequenceDTW.html
[5]https://numba.pydata.org/
[6]https://github.com/ARISE-Initiative/robomimic/tree/robocasa

A.6  QUALITATIVE RESULTS

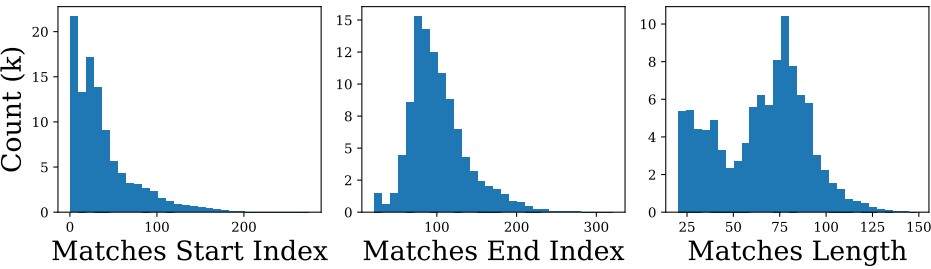

Figure 31: **Match distribution** $\mathcal{D}_{\mathbf{prior}}$ **for STRAP with target task:** *"put the black bowl in the bottom drawer of the cabinet and close it"*. S-DTW finds the best matches regardless of start and end points or trajectory length. This results in a distribution over start and end points as well as a variety of trajectory lengths retrieved.

