# OpenReview forum: "STRAP: Robot Sub-Trajectory Retrieval for Augmented Policy Learning"
_ICLR.cc/2025/Conference — ICLR 2025 Poster_

### Official Review · Reviewer_HWGB · 2024-10-19

**Soundness:** 2
**Presentation:** 1
**Contribution:** 3
**Rating:** 5
**Confidence:** 4

**Summary:**

## Paper Review Summary

This work advocates for training policies dynamically during deployment, utilizing the encountered scenarios to improve model performance. Instead of relying on pre-trained policies to tackle new problems in a zero-shot fashion, the authors propose a non-parametric approach that retrieves relevant data and trains models directly at test time. The paper introduces SRTAP, a method built on pre-trained Vision-Language Models (VLM) and dynamic time wrapping, which combines sub-trajectories into a policy. The approach involves some training on a set similar in language to the test set, and has been demonstrated in both real-world and simulated environments.

### Strengths:
1. **Innovative Approach**: The authors present a compelling intuition, demonstrating robustness in solving multitask generalization challenges.
2. **Efficient Data Usage**: The method shows improvement in the way data is leveraged for robotics tasks, particularly in sub-trajectory retrieval.
3. **Thorough Experiments**: The experiments are detailed and show promising results for sub-trajectory retrieval and policy creation.

### Weaknesses:
1. **Project Incompleteness**: There is no accessible website or supplementary information, suggesting the project might still be unfinished.
2. **Visual Readability**: The images in the paper are difficult to interpret, potentially detracting from the clarity of the results.
3. **Writing Quality**: The paper's writing needs improvement, especially in terms of clarity and readability.
4. **Generalization**: Some imitation learning methods, such as [Sparse Diffusion Policy](https://forrest-110.github.io/sparse_diffusion_policy/), [HPT](https://liruiw.github.io/hpt/), [RDT-Robotics](https://rdt-robotics.github.io/rdt-robotics), and [Humanoid Manipulation](https://humanoid-manipulation.github.io/), appear to show more generalization in similar settings.
5. **Formatting**: The paper's formatting is problematic, with certain sections being hard to read, affecting the overall readability of the work.

In conclusion, while the proposed method shows strong intuition and detailed experimentation, there are concerns about project completeness, readability, and potential improvements in both writing and generalization when compared to existing work.

I would like to change the rate after discussion, but at least you should finish the site you provide.

**Strengths:**

1. **Innovative Approach**: The authors present a compelling intuition, demonstrating robustness in solving multitask generalization challenges.
2. **Efficient Data Usage**: The method shows improvement in the way data is leveraged for robotics tasks, particularly in sub-trajectory retrieval.
3. **Thorough Experiments**: The experiments are detailed and show promising results for sub-trajectory retrieval and policy creation.

**Weaknesses:**

1. **Project Incompleteness**: There is no accessible website or supplementary information, suggesting the project might still be unfinished.
2. **Visual Readability**: The images in the paper are difficult to interpret, potentially detracting from the clarity of the results.
3. **Writing Quality**: The paper's writing needs improvement, especially in terms of clarity and readability.
4. **Generalization**: Some imitation learning methods, such as [Sparse Diffusion Policy](https://forrest-110.github.io/sparse_diffusion_policy/), [HPT](https://liruiw.github.io/hpt/), [RDT-Robotics](https://rdt-robotics.github.io/rdt-robotics), and [Humanoid Manipulation](https://humanoid-manipulation.github.io/), appear to show more generalization in similar settings.
5. **Formatting**: The paper's formatting is problematic, with certain sections being hard to read, affecting the overall readability of the work.

**Questions:**

1. Would you please update the sites in the paper?
2. Could you please add some experiments with other imitation learning methods?

---

> ### Author Response · Authors · 2024-11-23
>
> Dear Reviewer,
>
> Thank you for your feedback and for acknowledging our “compelling intuition” and “thorough experiments”.
>
> **Project incompleteness**
>
> We’ve updated the link to the website: https://strapaper.github.io/strap.github.io/ Furthermore, we’ve added more real-world experiments and improved baselines to the main paper and extended the appendix with experimental details and discussion on the computational complexity and choice of hyperparameters.
>
> **Visual readability and formatting**
>
> We acknowledge that some of the sections lacked proper formatting. We’ve improved the formatting by increasing the margins for figures, equations, and tables. Furthermore, we’ve updated [Figure 2] to be more interpretable and added pointers from the visual stages to the different sections in the paper. Could you please comment on whether these changes are sufficient and clarify what other changes to the writing quality and visual readability you would like to see?
>
> **Generalization to other imitation learning methods**
>
> We’ve run additional experiments using alternative architectures and imitation learning methods. Unfortunately, we found simpler architectures (MLP+GMM, LSTM+GMM) to struggle with achieving non-zero success rates on the LIBERO benchmark or limited by their capability to be conditioned on language (Diffusion Policies). While vision-language-action models experience much better language-conditioning, they require significantly more fine-tuning data than available in our setting [3]. Therefore, we chose transformer-based policies since they observe much better language-conditioning and multi-task capabilities as shown in [1,2].
>
> [1] Libero: Benchmarking knowledge transfer for lifelong robot learning. Liu et al, 2024
>
> [2] BAKU: An Efficient Transformer for Multi-Task Policy Learning. Haldar et al. 2024
>
> [3] OpenVLA: An Open-Source Vision-Language-Action Model. Kim et al. 2024
>
> We hope these additional modifications have addressed your previous questions. Please don’t hesitate to let us know if you have any additional comments or questions.

---

> > ### Comment · Reviewer_HWGB · 2024-11-24
> >
> > The first time I gave it a relatively low score, it was largely because the website had no content during my review, so I thought the article was in an unfinished stage.
> >
> > Although STARP has a certain degree of improvement in multi-task generalization ability compared with baseline, it usually requires few-shot learning. I don't think there is a significant advantage over the work of the same period, but I recognize the STARP work's contribution to the robotics community. I will improve my score, but only to 5 points.

---

> > > ### Author Response · Authors · 2024-11-25
> > > **Thank you for updating your score!**
> > >
> > > **Requiring few-shot learning**
> > >
> > > Deploying a multi-task baseline without any additional demonstrations achieves zero success on unseen tasks ($0.0$\% on 9/10 LIBERO-10 tasks). Collecting a small amount of demonstrations is therefore necessary to achieve any success at all. STRAP goes one step further and better uses the demonstration and available offline datasets to go beyond just success but to achieve robustness as demonstrated in our sim and real experiments.
> > >
> > > **Advantage over prior work**
> > >
> > > To touch on the advantages of STRAP over *work of the same period*, we point out that STRAP outperforms previous retrieval methods BehaviorRetrieval by $+12.2$\% and FlowRetrieval by $+12.5$\% on average across all LIBERO-10 tasks. With tuned hyperparameter $K$ this gap widens to $+24.7$\% and $+25.0$\%, respectively.
> > > In contrast to these baselines, STRAP does not require training an embedding model and instead uses an off-the-shelf frozen vision foundation model making it much more scalable to larger offline datasets.
> > >
> > > If this doesn’t fully address your question, we would greatly appreciate it if you could provide further clarification.

---

### Official Review · Reviewer_RToH · 2024-11-03

**Soundness:** 3
**Presentation:** 3
**Contribution:** 2
**Rating:** 6
**Confidence:** 4

**Summary:**

In this paper, the authors propose a task-specific robot learning framework using pre-collected datasets. Unlike the many robot learning methods that train the generalist policy model with multi-task expert data, the proposed method (STRAP) train a task-specific policies, which can yields better performance on single task. When the few-shot target demo, in addition to prior dataset is given, STRAP filters the task-relevant data from prior data and use it with target demo to train the model. One of the key features of STRAP is that it retrieves the data measuring the similarity between sub-trajectories, rather than the whole trajectories. Also, it utilize subsequence dynamic time warping (S-DTW) to match between the data. As a result, the proposed method shows improved performance compared to the previous methods, generalist policy models, and specialist models that only use the target data.

**Strengths:**

- To deal with potentially variable length during retrieval, STRAP use dynamic time warping (DTW) to match the sub-sequences
- STRAP shows improved performance compared to the prior framework (Behavior Retrieval, Du et al., 2023) which retrieves single state-action pairs using VAE.

**Weaknesses:**

- The idea of using only data relevant to the target task, rather than learning a generalist policy through multi-task data, is interesting. However, retrieving new data from prior dataset and training a policy each time a new scene is encountered is highly computationally costly.
- Comparing the entire prior data with the target data one-to-one to measure similarity is not scalable with the dataset size. Moreover, since this retrieval process requires computationally intensive neural network operations, such as DINO, it raises questions about whether this process can be performed at test time. In particular, there is no mention of how to handle an increase in offline dataset size, nor are there any discussions about limitations in this regard.
- There is no discussion about computational cost.
- STRAP uses a top-k retrieval dataset. Increasing this k could bring in more data but might reduce relevance, whereas a smaller k would provide more refined data but with a smaller amount. However, there is a lack of analysis on how changing this k value affects performance.

**Questions:**

- Compared to the prior framework (Behavior Retrieval, Du et al., 2023), STRAP seems to have three main differences in retrieval system. (a) use non-parametric retrieve vs. VAE (b) use sub-trajectory wise retrieve vs. single state-action pairs and (c) use DTW. Among these, what gives the most / least performance gains?

---

> ### Author Response · Authors · 2024-11-23
>
> Dear Reviewer,
>
> Thank you for your feedback on how we can improve our paper. We provide additional discussions on the computational complexity of our retrieval algorithm, an extensive ablation of hyperparameter $K$, and a detailed comparison to recent retrieval systems.
>
> **Computational cost to adapt to new scenes**
>
> Current generalist policies are unable to adapt zero-shot to the target task. A common approach is to fine-tune a pre-trained policy on a few target demonstrations (FT). While recent generalist policies suggest fine-tuning on 10-150 demonstrations [1] we only have access to 3-5 demonstrations in our few-shot setting. While the pre-training can provide faster adaptation, fine-tuning on a small number of demonstrations is unable to cover the randomization the policy is exposed to during evaluation (cf. added FT baselines in [Table 1,2 & 3]). By retrieving relevant data, we expose our expert to a much larger training distribution making it more robust to the deployment scenario.
>
> STRAP and a standard fine-tuning approach differ in the retrieval process and longer policy training. Our discussion in [A.5] shows that the retrieval scales linearly with the number of (sub-)trajectories in D_prior and D_target. Greater speedups can be achieved by parallelizing the retrieval and utilizing GPUs to compute the DTW distance matrix. Overall, STRAP’s runtime is longer compared to fine-tuning (\~5min) consisting of retrieval (\~5min on a DROID-scale dataset) and training a policy (\~30min) but provides significant robustness benefits shown by an average performance boost of $+6.4$% and $+25.7$% across all LIBERO-10 and real-world Kitchen tasks, respectively.
>
> [1] OpenVLA: An Open-Source Vision-Language-Action Model. Kim et al. 2024
>
> **Computational cost of the retrieval process**
>
> To avoid excessive compute during test time, we precompute the embeddings for D_prior! Encoding a single image running DINOv2 on an NVIDIA L40S 46GB and batch size 32 takes $2.83ms\pm 0.08$ (average across 25 trials). The wall clock time for encoding the entire DROID dataset (\~18.9M timesteps, single-view) therefore sums up to only \~26h. Every dataset only has to be encoded once when added to D_prior and can be reused for all future deployment scenarios. In contrast to previous methods (cf. BehaviorRetrieval, FlowRetrieval), using an off-the-shelf vision foundation model also eliminates the need to re-train the encoder and re-encode the entire dataset when D_prior grows.
>
> Retrieving data with S-DTW scales linearly with the size of D_prior, allowing for retrieval within \~5min even from the largest available datasets like DROID. Finally, STRAP’s policy learning stage is independent of the size of D_prior and only depends on the amount of retrieved data $K$, making it more scalable than common pre-training + fine-tuning or multi-task approaches that have to be re-trained when new trajectories are added to D_prior. We thoroughly discuss the complexity of our retrieval algorithm and benchmark the runtime of S-DTW in [A.5]. Note that there are several possible ways to improve our implementation, e.g., leveraging GPU deployment and custom CUDA kernels to compute the distance matrix or parallelizing retrieval across trajectories.

---

> ### Author Response · Authors · 2024-11-23
>
> **Choice of hyperparameter $K$**
>
> We extensively ablate STRAP for $K \in (100, 200, 500, 1000, 2000, 4000)$ on all LIBERO-10 tasks. We find tuning $K$ to improve our reported success rates ($K=100$)  on 8/10 tasks by an average of $7.5$%. The results suggest that the optimal value for $K$ is task-dependent with some tasks benefiting from retrieving less (4/10) and some from retrieving more (4/10) data. We hypothesize that the optimal $K$ depends on whether tasks benefit from broader less relevant or more refined data.
> Thank you for this suggestion! We update our results in [Tables 1 & 3] with the improved results.
> You can find additional details and visualizations in [Table 8] and [Figure 29] in the Appendix.
>
> The choice of $K$ has no impact on the computational complexity of STRAP as we always compute all matches between D_target and D_prior before selecting the top K. This also means that the hyperparameter search on $K$ can be reduced to policy learning and evaluation by storing the matches and training on $K$ segments.
>
>
> **Baselines ablations**
>
> a) We compare non-parametric (frozen DINOv2, denoted as D-S) and parametric (training a VAE as in Behavior Retrieval, denoted as BR) embeddings in a state-based retrieval setting. [Table 1] shows BR outperforming DINOv2 embeddings by $+5$% on average across all LIBERO-10 tasks. Inspecting the individual success rates in [Tables 1 & 3]  it becomes clear that the optimal embedding is highly task-dependent. However, we expect future representation learning techniques to close this gap. DINOv2 only encodes a single observation and, therefore, does not have a notion of dynamics and task semantics compared to BR.
>
> b) & c) We compare retrieving states, sub-trajectories, and full trajectory retrieval in [Table 1]. While state-based retrieval is based on cosine similarity, the variable length of trajectories requires S-DTW for sub-trajectory and full trajectory retrieval. We find retrieving sub-trajectories outperforms state-based retrieval by $+17.2%$ and full trajectory retrieval by $+4.2%$ on average across all LIBERO-10 tasks.
>
> To summarize, the largest gains come from retrieving (sub-)trajectories enabled by S-DTW. While the choice of embedding is largely task-dependent, STRAP allows for using off-the-shelf models by encoding the dynamics and semantics of the trajectories in the sub-trajectory retrieval process instead of the embedding model.
>
>
> We hope these additional modifications have addressed your previous questions. Please don’t hesitate to let us know if you have any additional comments or questions.

---

> > ### Comment · Reviewer_RToH · 2024-11-26
> >
> > Thank you for adressing my comments.
> > Even though the performance improvements seems promising, but the computation cost seems not to be practical.
> > If the retrieval takes ~5min in DROID dataset, does it mean that we need 5 min for interence for each action prediction?
> > if so, I would not agree that this algorithm is applicable to the real-world problems.

---

> > > ### Author Response · Authors · 2024-11-26
> > >
> > > Ah, there must be some misunderstanding! As a reminder, STRAP only runs retrieval once to augment the initial training dataset. We then train a Transformer-based policy on the augmented dataset [Section 4.5]. Afterward, the policy can run inference at 15Hz on the real robot (i.e., action prediction once every \~0.0667 seconds) [updated A.2]. The top of our website [[link](https://strapaper.github.io/strap.github.io/)] shows the real-time policy rollouts (no speedup). To summarize, STRAP does not run retrieval during action prediction but only a single time before training a policy.
> > >
> > > Please let us know if you have further questions!

---

> > > > ### Comment · Reviewer_RToH · 2024-11-26
> > > >
> > > > Thank you for adressing my questions!
> > > > I've updated the ratings.

---

### Official Review · Reviewer_JdfR · 2024-11-04

**Soundness:** 3
**Presentation:** 3
**Contribution:** 3
**Rating:** 8
**Confidence:** 4

**Summary:**

This paper focuses on the setting of generalizing a policy to an unseen task with few-shot demonstrations. Instead of deploying zero-shot, the paper proposes STRAP, training a model on task-relevant data augmented by retrieval. STRAP first retrieves sub-trajectories from a large pretraining dataset that are similar to the new task demonstrations, then combines them with the few-shot demos to train a policy. Results on sim and real environments show that STRAP outperforms other retrieval methods and pure behavioral cloning. Ablations show that STRAP is compatible with various vision encoders and justify each of its component.

**Strengths:**

- Sub-trajectory retrieval for the few-shot demo behavioral cloning setting is a well-motivated and novel idea.
- The method is clear and straightforward to implement.
- Results show that matching with DTW on vision foundation model features are robust to variations and capture task semantics.
- Real and simulated environments show that STRAP outperforms other retrieval methods and pure behavioral cloning.

**Weaknesses:**

- STRAP requires few-shot demos and model training at test time for a new task.
- It would be good to see more sim and real environments for evaluations.
- It would be more convincing to see a behavioral cloning baseline that uses all available data.

**Questions:**

- Baseline: how does STRAP compare to the pretrain-then-finetune setup? (Pretrain on the prior dataset, then fine-tune on the few-shot target demonstrations?)
- Baseline: how does STRAP compare to a multitask policy trained on all available data?
- Generalization: to what extent does STRAP generalize across environments or embodiments?

---

> ### Author Response · Authors · 2024-11-23
>
> Dear Reviewer,
>
> Thank you for your feedback on how we can improve our paper. We provide additional results on improved baselines, discussions on the runtime differences between STRAP and a fine-tuning approach, and generalization to new environments and embodiment.
>
> **Additional real-world tasks**
>
> We’ve added three real-world tasks (“pick and place the [carrot, pepper, chili]”) in three realistic kitchen environments ([table, sink, stove]) and collected 50 multi-task demonstrations in every scene. Object poses are randomized in a $20\times20cm$ grid during data collection and evaluation. D_target is specific to each environment and contains 3 demonstrations of the downstream task.
>
> We find STRAP to experience surprising generalization behavior from the 3 poses seen in D_target to the poses in the $20\times20cm$ grid exposed during evaluation (Kitchen). The policy further shows recovery behavior, completing the task even when the initial grasp fails and alters the object pose [see the robot attempting to pick up the red pepper in the first video on our website]. As shown in [Table 2] STRAP even maintains its high performance, retrieving the relevant data for much larger offline datasets (Kitchen+DROID).
>
> Meanwhile, the large pose randomizations in D_prior and the evaluation are challenging for the baselines. Behavior cloning and fine-tuning (BC, FT) fit the target demonstrations failing to adapt to unseen object poses. The multi-task policy (MT) replays trajectories from the offline dataset instead of solving the prompted task most likely caused by an imbalanced training dataset. Increasing the dataset size amplifies these challenges by making it significantly harder to pre-train or learn a multi-task policy, leading to performance drops of $-6.09$% (FT) and $-26.3$% (MT).
>
> **Stronger baselines**
>
> We extend our evaluations by including two additional baselines for all LIBERO-10 tasks and the real-world tasks introduced above.
>
> 1) Pre-training + fine-tuning (FT) which represents a policy pre-trained on D_prior and fine-tuned on the few available demonstrations in D_target.
> 2) Multi-task policy (MT) trained on D_prior $\cup$ D_target in contrast to only D_prior in our original submission.
>
> The fine-tuning baseline is more competitive than standard behavior cloning but still falls short by $-6.4$% and $-25.7$% compared to STRAP on the LIBERO-10 and real-world tasks. The updated multi-task policy performs well on some tasks where we expect positive transfer, i.e., where environments and tasks in D_prior and D_target overlap, e.g., LIBERO-10 and LIBERO-90 both contain the Book-Caddy task. We’ve added the FT baseline and replaced MT trained only on D_prior with MT trained on D_prior $\cup$ D_target in [Tables 1,2 & 3].

---

> > ### Comment · Reviewer_JdfR · 2024-11-27
> > **Thank you for the additional experiments and insights**
> >
> > Thank you for the additional experiments and insights. From the additional baselines, it seems that subtrajectory retrieval outperforms multitask behavioral cloning and fine-tuning, given access to the same pool of data. Additional experiments also show that STRAP can retrieve semantically relevant subtrajectories from large datasets, and improves robustness in the few-shot setting. My concerns have been addressed and I have updated my rating.

---

> ### Author Response · Authors · 2024-11-23
>
> **Few-shot demos and model training at test time**
>
> Current generalist policies are unable to adapt zero-shot to the target task. A common approach is to fine-tune a pre-trained policy on few target demonstrations (FT). While recent generalist policies suggest fine-tuning on 10-150 demonstrations [1] we only have access to 3-5 demonstrations in our few-shot setting. While the pre-training can provide faster adaptation, fine-tuning on 3 demonstrations is unable to cover the $20\times20cm$ grid of possible object poses the policy is exposed during evaluation (cf. added FT baselines in [Table 1,2 & 3]). By retrieving relevant data, we expose our expert to a much larger training distribution making it more robust to the deployment scenario.
>
> STRAP and a standard fine-tuning approach differ in the retrieval process and longer policy training. Our discussion in [A.5] shows that the retrieval scales linearly with the number of (sub-)trajectories in D_prior and D_target. Greater speedups can be achieved by parallelizing the retrieval and utilizing GPUs to compute the DTW distance matrix. Overall, STRAP’s runtime is longer compared to fine-tuning (\~5min) consisting of retrieval (\~5min on a DROID-scale dataset) and training a policy from scratch (\~30min) but provides significant robustness benefits shown by an average performance boost of $+6.4$% and $+25.7$% across all LIBERO-10 and real-world Kitchen tasks, respectively.
>
> [1] OpenVLA: An Open-Source Vision-Language-Action Model. Kim et al. 2024
>
>
> **Cross-environment and -embodiment generalization**
>
> We provide examples of sub-trajectories retrieved from the DROID dataset on our website [see “Does retrieval scale to DROID?”]. STRAP retrieves trajectories collected in environments with similar appearance, e.g., camera pose, table orientation, and texture, and similar tasks, e.g., picking up cylindrical objects. The choice of embedding allows for further steerability of the retrieved sub-trajectories. For instance, averaging the embeddings of all three cameras leads to retrieval of very similar scenes, while using embeddings of only the in-hand camera focuses much more on the manipulated object and can be embodiment-agnostic. While the retrieval could scale to multiple embodiments, training cross-embodied policies is much more challenging, and leveraging cross-embodied data is an open research problem that we leave to future work.
>
> We hope these additional modifications have addressed your previous questions. Please don’t hesitate to let us know if you have any additional comments or questions.

---

> ### Author Response · Authors · 2024-11-27
>
> We hope our responses have adequately addressed your concerns! If so, we kindly ask you to increase your score recommendation.

---

### Official Review · Reviewer_Ndo3 · 2024-11-04

**Soundness:** 2
**Presentation:** 3
**Contribution:** 3
**Rating:** 6
**Confidence:** 4

**Summary:**

The paper introduces STRAP, a novel method for trajectory retrieval to find similar sub-trajectory segments in large-scale datasets for efficient policy learning in few-shot settings. The method's key contribution lies in combining pretrained visual encoders with Dynamic Time Warping to encode sub-trajectories of variable lengths for improved retrieval. The proposed method is tested against several retrieval baselines and BC ones on LIBERO-10 simulation and real world pick and place tasks and achieves good performance.

**Strengths:**

- The proposed method is well motivated and achieves strong results across multiple experiments, both in simulation and real-world settings

- The use of Dynamic Time Warping for sub-trajectory matching is novel and well-suited for the problem domain

- Comprehensive evaluation against recent retrieval baselines demonstrates the method's effectiveness

- Thorough ablation studies on different pretrained encoders provide valuable insights into architecture choices

- The paper is well written and includes several illustrative figures that enhance the text

**Weaknesses:**

- The baseline comparison against multi-task policy appears weak, as it only uses pretrained weights without fine-tuning. This seems like an artificially weak baseline since fine-tuning is standard practice for all MT-policies.

- The paper's argument that retrieval is more efficient than expensive pretraining needs stronger empirical support, especially given that the robotics community regularly fine-tunes general policies for downstream tasks

- The computational cost of STRAP's retrieval process on large-scale datasets like Droid is not adequately addressed, raising questions about real-world scalability. Some more clarity is necessary here

- The choice of K for constructing D-retrieval lacks sufficient explanation and ablations. The paper should explore how different K values affect both retrieval quality and computational overhead and policy performance, as this parameter likely presents a trade-off between performance and efficiency. A discussion about the retrieved data quantity would provide valuable insights and strengthen then paper.

- Small number of tested tasks in real world setting and missing baselines of MT policy and finetuned MT policy

**Questions:**

- Could STRAP be combined with fine-tuning of MT policies on the retrieved dataset to potentially achieve even better performance than domain-specific fine-tuning alone?

- How does STRAP's performance compare against standard fine-tuning approaches when controlling for the total amount of data used?

- What are the memory and computational requirements for deploying STRAP on very large trajectory datasets like Droid?

- Can you add the average performance for LIBERO-10 results to the main table?

- Can you provide a few more real world tasks with required baselines?

- Real world retrieveal is conducted with the same robot embodiment and gripper. How does STRAP Perform when retrieving similar data from other robot datasets like BridgeV2 that does not share the same robot and scenes?

STRAP presents an novel and well designed approach to few-shot learning, that tackles several drawbacks of prior methods through sub-trajectory retrieval with dynamic time-warping. However, more comprehensive comparisons against fine-tuned baselines and clearer analysis of computational requirements would strengthen the paper's contributions. Thus, I recommend weak reject pending addressing the following concerns: (1) comparisons against fine-tuned baselines, (2) clearer analysis of computational requirements, and (3) better justification of parameter choices.

---

> ### Author Response · Authors · 2024-11-23
>
> Dear Reviewer,
>
> Thank you for your feedback on how we can improve our paper. We provide additional results on three real-world tasks, improved baselines, an extensive ablation of hyperparameter $K$, and a discussion on our retrieval algorithm's computational and memory complexity.
>
> **Additional real-world tasks**
>
> We’ve added three real-world tasks (“pick and place the [carrot, pepper, chili]”) in three realistic kitchen environments ([table, sink, stove]) and collected 50 multi-task demonstrations in every scene. Object poses are randomized in a $20\times20cm$ grid during data collection and evaluation. D_target is specific to each environment and contains 3 demonstrations of the downstream task.
>
> We find STRAP to experience surprising generalization behavior from the 3 poses seen in D_target to the poses in the $20\times20cm$ grid exposed during evaluation (Kitchen). The policy further shows recovery behavior, completing the task even when the initial grasp fails and alters the object pose [cf. robot attempting to pick up the red pepper in the first video on our website]. As shown in [Table 2] STRAP even maintains its high performance, retrieving the relevant data for much larger offline datasets (Kitchen+DROID).
>
> Meanwhile, the large pose randomizations in D_prior and the evaluation are challenging for the baselines. Behavior cloning and fine-tuning (BC, FT) fit the target demonstrations failing to adapt to unseen object poses. The multi-task policy (MT) replays trajectories from the offline dataset instead of solving the prompted task most likely caused by an imbalanced training dataset. Increasing the dataset size amplifies these challenges by making it significantly harder to pre-train or learn a multi-task policy, leading to performance drops of $-6.09$% (FT) and $-26.3$% (MT).
>
>
>
> **Stronger baselines**
>
> We extend our evaluations by including two additional baselines for all LIBERO-10 tasks and the real-world tasks introduced above.
>
> 1) Pre-training + fine-tuning (FT) which represents a policy pre-trained on D_prior and fine-tuned on the few available demonstrations in D_target.
> 2) Multi-task policy (MT) trained on D_prior $\cup$ D_target in contrast to only D_prior in our original submission.
>
> The fine-tuning baseline is more competitive than standard behavior cloning but still falls short by $-6.4$% and $-25.7$% compared to STRAP on the LIBERO-10 and real-world tasks. The updated multi-task policy performs well on some tasks where we expect positive transfer, i.e., where environments and tasks in D_prior and D_target overlap, e.g., LIBERO-10 and LIBERO-90 both contain the Book-Caddy task. We’ve added the FT baseline and replaced MT trained only on D_prior with MT trained on D_prior $\cup$ D_target in [Tables 1,2 & 3].

---

> ### Author Response · Authors · 2024-11-23
>
> **Computational cost, memory, and scalability to real-world datasets**
>
> STRAP’s retrieval stage is based on S-DTW which consists of two stages: computing the distance matrix $D$ and finding the shortest path via dynamic programming.
> These stages have to be run sequentially for each sub-trajectory in D_target and trajectory in D_prior but don’t depend on the other (sub-)trajectories. Therefore, STRAP has a runtime complexity of $\mathcal{O}(N*M)$ with N the number of sub-trajectories in D_target and M the number of trajectories in D_prior.
>
> We thoroughly benchmark the runtime of S-DTW in [A.5]. For an offline dataset the size of
> DROID (76k), retrieval takes approximately $300sec$ using our unoptimized research code. Note that there are several possible ways to improve our implementation, e.g., leveraging GPU deployment and custom CUDA kernels to compute the distance matrix or parallelizing retrieval across trajectories.
>
> Since we encode D_prior using vision foundation models, the memory footprint of STRAP stays fairly low. Loading the embeddings to compute the cost matrix represents the largest memory consumption but is much lower than the memory consumed by loading image or video sequences.
>
> For runtime visualization and information regarding data encoding and policy learning complexity please refer to [A.5] in the appendix.
>
> **Choice of hyperparameter $K$**
>
> Thank you for this suggestion! We extensively ablate STRAP for $K \in (100, 200, 500, 1000, 2000, 4000)$ on all LIBERO-10 tasks. We find tuning $K$ to improve our previously reported success rates ($K=100$)  on 8/10 tasks by an average of $7.5$%. The results suggest that the optimal value for $K$ is task-dependent with some tasks benefiting from retrieving less (4/10) and some from retrieving more (4/10) data. We hypothesize that the optimal $K$ depends on whether tasks leverage (positive transfer) or suffer (negative transfer) from multi-task training. We update our results in [Tables 1 & 3] with the improved results. You can find additional details and visualizations in [Table 8] and [Figure 29] in the Appendix.
>
> The choice of $K$ has no impact on the computational complexity of STRAP as we always compute all matches between D_target and D_prior before selecting the top $K$. This also means that the hyperparameter search on $K$ can be reduced to policy learning and evaluation by storing the matches and retrieving $K$ segments for each iteration.
>
> **Cross-embodiment generalization**
>
> We leave cross-embodiment retrieval to future work but emphasize that the embedding choice allows for some steerability of the retrieved sub-trajectories. For instance, averaging the embeddings of all three cameras leads to retrieving very similar scenes [see “Does retrieval scale to DROID?” on our website] while using embeddings of only the in-hand camera focuses much more on the manipulated object and can be embodiment-agnostic. While the retrieval might scale to multiple embodiments, training cross-embodied policies is much more challenging, and leveraging cross-embodied data is an open research problem.
>
>
>
> We hope these additional modifications have addressed your previous questions. Please don’t hesitate to let us know if you have any additional comments or questions.

---

> > ### Comment · Reviewer_Ndo3 · 2024-11-25
> >
> > Thank you for your efforts in the rebuttal. The new experiments and baselines address my initial concerns. Thus, I raise my score to 6.
> >
> > I got one follow-up question:
> >
> > How does FT with Strap retrieved trajectories do on LIBERO-10 and real world setting?

---

### Author Response · Authors · 2024-11-23
**Thank you for your feedback!**

We thank all reviewers for their constructive feedback and for acknowledging the novelty of Dynamic Time Warping for sub-trajectory retrieval [Ndo3, JdfR, RToH], pointing out its improvements over recent methods [Ndo3, JdfR, RToH], and describing our intuition as well motivation and compelling [Ndo3, JdfR, HWGB].


We’ve added three new real-world tasks (“pick and place the [carrot, pepper, chili]”) to our evaluation (Kitchen) and improved the baselines (in sim and real) by adding pre-training + fine-tuning (FT) and a multi-task policy trained on all available data (MT).

Kitchen
|           | Table | Sink | Stove |
|----------------|---------|--------------|--------------|
| BC | 12.50   | 10.00         | 14.28         |
| FT   | 20.00    | 27.27         | 30.43         |
| MT     | 4.34    | 31.57         | 45.00         |
| STRAP     | **36.36**    | **61.36**         | **57.12**         |

To investigate scalability to larger datasets, we construct an additional offline dataset D_prior consisting of 5000 demonstrations from the DROID dataset and 50 demonstrations collected in the same environment as D_target (Kitchen+DROID).

Kitchen+DROID
|           | Table | Sink | Stove |
|----------------|---------|--------------|--------------|
| BC | 12.50   | 10.00         | 14.28         |
| FT   | 28.00    | 8.69         | 22.72         |
| MT     | 2.00    | 0.00        | 0.00         |
| STRAP     | **56.81**    | **63.04**         | **45.45**         |


We address your feedback in the comment section below. You can find extensive ablations of hyperparameter $K$ [A.3] and a discussion on the computational complexity of STRAP [A.5] in the appendix. We’ve updated the manuscript for better readability and included additional real-world experiments and improved baselines.

---

### Meta-Review · Area_Chair_fCkJ · 2024-12-17

**Metareview:**

(a) This paper propose a novel few-shot imitation learning approach, STRAP, based on sub-trajectory retrieval. In simulated and real-world experiments, STRAP outperformed existing retrieval algorithms and multi-task learning methods, demonstrating its scalability to large offline datasets and its ability to learn robust control policies from limited real-world demonstrations.

(b) Strengths:
- Novelty: Introducing sub-trajectory retrieval is a novel contribution.
Effectiveness: STRAP demonstrates strong performance improvements over baseline methods in both simulated and real-world experiments.
Clarity: The paper is generally well-written and includes illustrative figures that enhance understanding.

(c) Weakness:
- Computational Cost: While the authors responded to the computational cost of retrieval (Sec A.5), it still remains a concern to some extent.
- It'd make the paper stronger to have a deeper investigation of generalization.

(d) My decision is to accept the paper for the novelty and strong performance.

**Additional Comments On Reviewer Discussion:**

The reviewers generally agree that STRAP is well-motivated and novel for few-shot imitation learning. They generally appreciate the clear presentation, thorough experiments, and promising results in both simulated and real-world settings.

The reviewers initially raised concerns regarding the strength of multi-task policy baselines, choice of hyperparameter K; questions regarding memory and computational requirements.

The authors responded to the reviewers' feedback and addressed most concerns in the rebuttal. They have added additional new real-world tasks to their evaluation, demonstrating STRAP's robustness and generalization capabilities; included two additional baselines, pre-training with fine-tuning (FT) and a multi-task policy trained on all available data (MT), strengthening the comparisons; addressed questions about scalability; conducted ablations of the hyperparameter K.

These efforts led to improvement of the review scores, moving the paper from borderline (positively) to accept.

---

### Decision · Program_Chairs · 2025-01-22

Accept (Poster)